# TRIDE: A Text-assisted Radar-Image weather-aware fusion network for Depth Estimation

**Huawei Sun**                                                                              *huawei.sun@tum.de*
*Technical University of Munich*
*Infineon Technologies AG*

**Zixu Wang**                                                                                *zixu.wang@tum.de*
*Technical University of Munich*
*Infineon Technologies AG*

**Hao Feng**                                                                                  *hao.feng@tum.de*
*Technical University of Munich*

**Julius Ott**                                                                                *julius.ott@tum.de*
*Technical University of Munich*
*Infineon Technologies AG*

**Lorenzo Servadei**                                                                    *lorenzo.servadei@tum.de*
*Technical University of Munich*

**Robert Wille**                                                                            *robert.wille@tum.de*
*Technical University of Munich*

**Reviewed on OpenReview:** *https://openreview.net/forum?id=ZMqMnwMfse*

## Abstract

Depth estimation, essential for autonomous driving, seeks to interpret the 3D environment surrounding vehicles. The development of radar sensors, known for their cost-efficiency and robustness, has spurred interest in radar-camera fusion-based solutions. However, existing algorithms fuse features from these modalities without accounting for weather conditions, despite radars being known to be more robust than cameras under adverse weather. Additionally, while Vision-Language models have seen rapid advancement, utilizing language descriptions alongside other modalities for depth estimation remains an open challenge. This paper first introduces a text-generation strategy along with feature extraction and fusion techniques that can assist monocular depth estimation pipelines, leading to improved accuracy across different algorithms on the KITTI dataset. Building on this, we propose TRIDE, a radar-camera fusion algorithm that enhances text feature extraction by incorporating radar point information. To address the impact of weather on sensor performance, we introduce a weather-aware fusion block that adaptively adjusts radar weighting based on current weather conditions. Our method, benchmarked on the nuScenes dataset, demonstrates performance gains over the state-of-the-art, achieving a 12.87% improvement in MAE and a 9.08% improvement in RMSE. Code: `https://github.com/harborsarah/TRIDE`

## 1 Introduction

Estimating a dense depth map for a given scene is essential for 3D scene reconstruction, which can be achieved using either an RGB image alone or a combination of an RGB image and a sparse depth map. With advancements in deep learning, learning-based monocular depth estimation methods Lee et al. (2019); Li et al.; Shao et al. (2024); Patil et al. (2022); Agarwal & Arora (2023); Liu et al. (2023); Piccinelli et al.

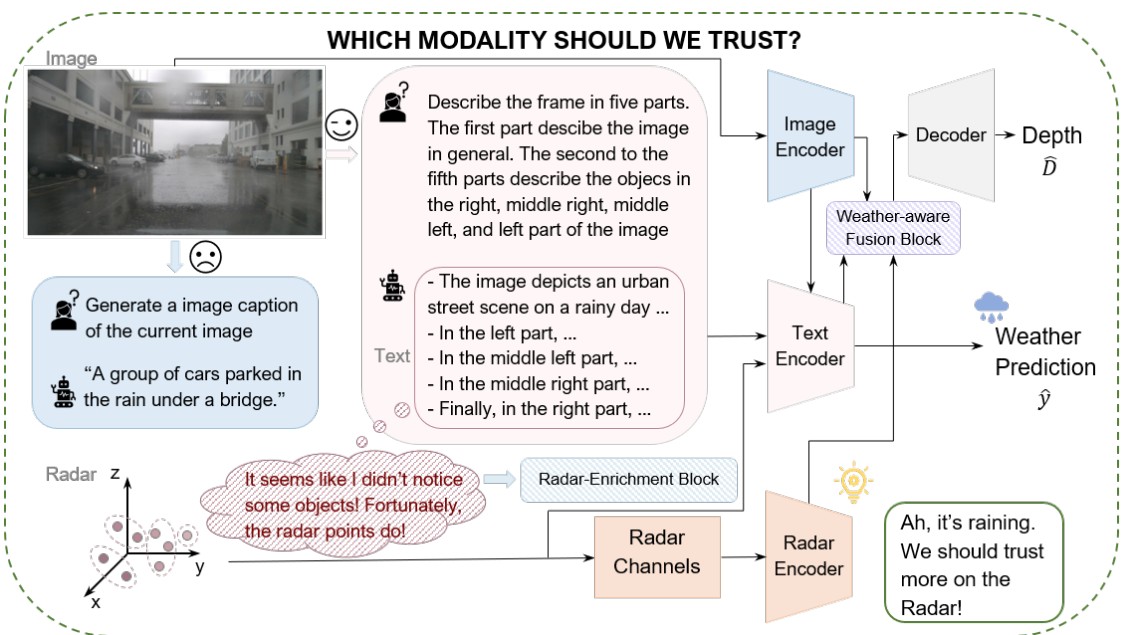

Figure 1: **Which modality should we trust?** Our proposed text generation approach generates detailed general and regional descriptions from the image. However, both image and language descriptions are susceptible to adverse weather conditions, a limitation that can be addressed by the robustness of radar.

(2024); Yuan et al. (2022); Zeng et al. (2024a) have significantly outperformed traditional approaches Saxena et al. (2005; 2008); Wang et al. (2015) in terms of accuracy. To address the inherent challenges of this ill-posed problem, many studies Yin et al. (2019); Qi et al. (2018); Liu et al. (2023); Lee et al. (2019) incorporate geometric cues and scene priors to enhance depth accuracy. For instance, WorDepth Zeng et al. (2024a) leverages language as a prior to enhance the depth estimation process. However, as it generates only a single caption per frame, the impact of incorporating language is limited, particularly in complex outdoor scenarios.

In the realm of depth completion, LiDAR-camera depth completion Hu et al. (2021); Cheng et al. (2020); Tang et al. (2024); Wang et al. (2023b); Yan et al. (2024; 2023); Park et al. (2020) has achieved high performance by effectively processing irregularly sampled sparse depth points from LiDAR sensors. Recently, radar-camera depth estimation has gained popularity as a potential alternative to LiDAR, as radar sensors are cost-effective and perform robustly in adverse weather conditions. However, radar point clouds are substantially sparser and noisier than LiDAR, posing additional challenges. Current research Nobis et al. (2019); Sun et al. (2023; 2024a); Long et al. (2021); Singh et al. (2023); Sun et al. (2024b; 2025c;a) has focused primarily on addressing the sparsity and noise issues in radar data, with less emphasis on exploring more effective fusion strategies to capitalize on the strengths of different sensors. With fusion methods proposed in Singh et al. (2023); Sun et al. (2024b), the model is still not able to adjust the weight between image and radar features under different weather conditions. Given that radar sensors are particularly robust under adverse weather conditions, incorporating weather information could allow the radar branch to be more heavily weighted in challenging environments.

To address the abovementioned limitations, we begin by exploring how to fully leverage language information in estimating depth. We utilize the capabilities of rapidly advancing Multimodal Large Language Models (MLLMs) to generate detailed descriptions of each frame. Specifically, we propose a text-generation strategy that describes the image in two scopes: a general overview and region-specific details. During decoding, these text features are integrated into the model through general and regional attention blocks at two stages, effectively enhancing the decoding process. To demonstrate the impact of integrating language information on depth estimation, we first conduct experiments on the KITTI benchmark Geiger et al. (2012). By incorporating the text branch into several existing monocular depth estimation algorithms, our approach achieves superior performance compared to the original models.

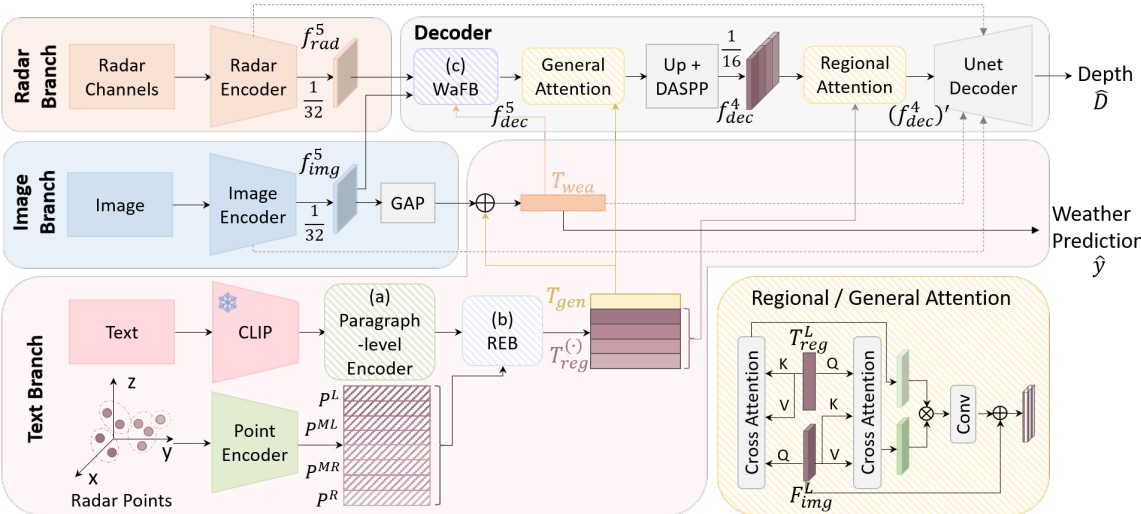

Figure 2: Model Architecture. Our model consists of four parts. Radar, image, and text branches encode features individually and send them into the decoder, outputting the final depth map. The text branch also predicts the weather of the current frame to ensure extracting reasonable weather-aware feature for the fusion process.

Language descriptions generated from images are often impacted by adverse weather conditions. For instance, in night-time or rainy scenarios, images can become blurry or obscured by water drops, reducing the quality of the generated text descriptions. This leads to the question: Which modality should we trust to estimate depth more accurately? Figure. 1 illustrates the overall concept of our proposed method, showing that incorporating text descriptions into a radar-camera system can yield further improvements, as radar sensors remain robust under such challenging conditions. In this context, we introduce TRIDE, a multi-modal depth estimation framework that integrates information from radar, camera, and language to enhance robustness. To correct and refine the text information during feature extraction, we propose a Radar-Enrichment Block (REB) that incorporates region-specific 3D radar point information into the text features. Additionally, we introduce a Weather-aware Fusion Block (WaFB) that adaptively weights the radar branch based on weather information extracted from image and text, effectively guiding the fusion of radar and image features. Text features are integrated into the decoder in a way that is consistent with the monocular depth estimation framework. We train and evaluate our approach on the nuScenes Caesar et al. (2020) dataset, a widely recognized benchmark for radar-camera-based solutions, where it achieves state-of-the-art performance compared to existing radar-camera depth estimation algorithms. To summarize, our contributions are as follows:

- We introduce a novel text branch, incorporating text-generation and feature extraction strategies, along with general and regional attention blocks to effectively integrate language information into image features. Integrating this text branch into existing monocular depth estimation algorithms improves depth estimation accuracy on the KITTI dataset.

- We propose TRIDE, a multi-modal framework that surpasses existing radar-camera depth estimation algorithms on the nuScenes dataset.

- To maximize the robustness advantages provided by the radar sensor, we first propose REB to enrich the regional text features with robust radar point features. Then, we design a novel fusion block, WaFB, that leverages weather information during fusion.

To the best of our knowledge, this is the first work to leverage detailed language descriptions to enhance radar-camera depth estimation while accounting for weather conditions during the fusion process.

## 2 Related Work

**Language Models for Autonomous Driving.** Vision-Language Models (VLMs) Li et al. (2022; 2023); Radford et al. (2021); Oquab et al. (2023); Zhu et al. (2023) are designed to learn rich vision-language correlations from diverse datasets, enabling zero-shot predictions in downstream tasks. Despite their potential, leveraging language as an auxiliary modality to enhance performance in Autonomous Driving (AD) remains an under-explored area. Some recent works have begun investigating this potential in different AD tasks. The system in Kim et al. (2020) learns to predict action responses by summarizing visual observations in natural language. Keysan *et al.* Keysan et al. (2023) introduced text-based representations to improve trajectory prediction. In 3D localization, several studies Kolmet et al. (2022); Xia et al. (2024); Wang et al. (2023a) focus on matching text descriptions with 3D point clouds. DRAMA Malla et al. (2023) aims to localize risk objects using free-form language descriptions.

For depth estimation, only a few researchers have explored the integration of language descriptions. WorDepth Zeng et al. (2024a) learned the distribution of 3D scenes from text captions, while RSA Zeng et al. (2024b) predicts scale from language, providing an explicit scaling constraint by transforming relative depth into metric scale. However, these approaches rely on a single caption per image, often insufficient for describing complex outdoor driving scenes. Thus, generating meaningful language descriptions and effectively extracting useful text information for complex scenarios is still challenging.

**Radar-Camera Depth Estimation.** Radar data offers valuable depth priors in adverse weather conditions, but radar-camera depth estimation presents significant challenges due to the inherent sparsity and noise in radar data. Earlier approaches addressed the sparsity issue by reprojecting radar points from multiple sweeps onto the current frame Lin et al. (2020); Lo & Vandewalle (2021); Long et al. (2021). Recently, the focus has shifted toward leveraging single-scan radar data Singh et al. (2023); Sun et al. (2024b); Li et al. (2024a); Sun et al. (2025c); Li et al. (2024b); Sun et al. (2025b) for better real-time performance. To mitigate sparsity, several methods Li et al. (2024a); Singh et al. (2023); Sun et al. (2024b) employ two-stage architectures that first estimate a quasi-dense depth map, followed by the final depth estimation in a second stage. While GET-UP Sun et al. (2025c) completes depth estimation within one stage, which incorporates point cloud upsampling as an auxiliary task to avoid introducing extraneous noise in the densification process. Besides addressing noise and sparsity issues, effectively fusing different modalities is equally important. Most existing approaches Lin et al. (2020); Lo & Vandewalle (2021); Li et al. (2024b) perform information aggregation by simply concatenating radar and image features. However, this straightforward concatenation may lead to ineffective convolution over many zero activations due to the sparse nature of radar features. To address this, gated fusion was proposed in Singh et al. (2023) to more effectively leverage radar information. As an enhancement, CaFNet Sun et al. (2024b) introduces confidence-aware gated fusion, incorporating a predicted radar confidence map to ignore regions with noisy radar data. Cross-attention is used in Li et al. (2024a) to integrate radar information into the image features, which helps to focus on critical information from each modality.

Despite these advancements, existing approaches do not consider weather information during fusion. Moreover, how to effectively incorporate text information to enrich radar-camera fusion-based algorithms remains an open question.

## 3 Approach

This section begins by presenting our text-generation method, followed by an overview of the proposed TRIDE architecture. Next, we introduce the text encoding and decoder structures. Subsequently, we discuss the loss functions used for training TRIDE. Finally, we explain how text can assist in monocular depth estimation.

### 3.1 Text Generation

To fully leverage the power of text-based information, rather than using image captioning models, we generate a detailed description of the input image by using a specially designed prompt as input to the model MiniCPM-V Yao et al. (2024). The MLLM, denoted as $\mathcal{M}$, takes an image $x_{\text{img}}$ and a prompt $p$ as inputs, and outputs five paragraphs: The first paragraph $\mathbf{t}_{\text{gen}}$ provides an overview of the image, including details

such as the current weather conditions. The remaining four paragraphs provide region-specific descriptions. Here the image is divided into four regions: left (L), middle-left (ML), middle-right (MR), and right (R), with descriptions $\mathbf{t}_{\text{reg}}^{\text{L}}, \mathbf{t}_{\text{reg}}^{\text{ML}}, \mathbf{t}_{\text{reg}}^{\text{MR}}, \mathbf{t}_{\text{reg}}^{\text{R}}$, each containing $n_{(\cdot)}$ sentences, respectively. The descriptions for each region contain information about the objects present and their estimated depth within the scene. Therefore, language descriptions can serve as priors, providing additional information to support depth estimation. Additionally, region-specific descriptions offer more detailed insights into the image, allowing for easier alignment and fusion with image features. The designed prompt and an example of the text description are shown in the supplementary material.

## 3.2 Model Architecture

Our radar-camera fusion model, illustrated in Figure 2, processes information through three main branches. The image branch uses a ResNet-34 encoder He et al. (2016) to process an input image $x_{\text{img}} \in \mathbb{R}^{H \times W \times 3}$, generating five multi-scale features $\{f_{\text{img}}^i\}_{i=1}^5$, where $H$ and $W$ denote the image's height and width, respectively. For radar information, given a radar point cloud $\mathbf{r} \in \mathbb{R}^{N \times C_r}$, we project the radar points onto the image plane, forming radar projection channels, $x_{\text{rad}} \in \mathbb{R}^{H \times W \times (C_r - 2)}$. Here, $N$ and $C_r$ represent the number of radar points and the features per point, respectively. These channels contain information such as depth, velocity, and Radar Cross Section (RCS), which are processed by a ResNet-18 encoder He et al. (2016) to extract 2D radar features.

The text branch processes the extracted text and radar point cloud $\mathbf{r}$ to produce a general text feature $T_{\text{gen}}$ and four region-specific text features, $T_{\text{reg}}^{\text{L}}, T_{\text{reg}}^{\text{ML}}, T_{\text{reg}}^{\text{MR}}$, and $T_{\text{reg}}^{\text{R}}$. Each text feature is a vector of dimension $C_t$. Moreover, a weather-aware feature $T_{\text{wea}}$ is generated by combining the pooled final image feature $f_{\text{img}}^5$ with $T_{\text{gen}}$, enhancing the model's capability to predict weather-related information.

In the decoding phase, we utilize a UNet Ronneberger et al. (2015) structure. Image and radar features are fused through the Weather-aware Fusion Block (WaFB), which incorporates the weather feature $T_{\text{wea}}$. Additionally, text features are integrated at two stages: $T_{\text{gen}}$ is fused at a scale of $\frac{1}{32}$, while the regional features apply regional attention with the feature at a scale of $\frac{1}{16}$. Further details are provided in Sec. 3.4.

## 3.3 Text Encoding

The text branch processes the extracted text alongside the radar point cloud $\mathbf{r} \in \mathbb{R}^{N \times C_r}$, where $N$ represents the number of radar points in the current frame, and $C_r$ denotes the size of the radar point features. The goal is twofold. First, the text sometimes contains roughly estimated depth information generated by the MLLM, which can be noisy and inaccurate. Integrating radar point information into the text features helps rectify this inaccuracy. Second, in adverse weather conditions, images can appear blurred, leading to even noisier information in the text. By incorporating robust radar point information, the text feature extraction process is better guided, improving alignment with the correct spatial positions.

To encode the textual information, we first employ the text encoder from the pre-trained vision-language model CLIP Radford et al. (2021) to capture *sentence-level* features within each paragraph. For example, given a paragraph $\mathbf{t}_{\text{reg}}^{\text{L}}$ containing $n_{\text{L}}$ sentences, CLIP encodes the features as $\mathbf{f}^{\text{L}} = \{f_1^{\text{L}}, f_2^{\text{L}}, \ldots, f_{n_{\text{L}}}^{\text{L}}\}$, with each feature vector $f_i^{\text{L}} \in \mathbb{R}^C$.

To manage the variation in sentence counts across paragraphs and to extract a unified feature representation for each paragraph, we introduce a *paragraph-level* encoder block, illustrated in Figure 3. This block employs Long Short-Term Memory (LSTM) layers Hochreiter (1997) to generate a global feature representation for each paragraph, leveraging LSTM's ability to capture long-term dependencies. By sequentially processing the sentence features, the LSTM extracts a comprehensive feature representation that encapsulates the primary information of each paragraph. After passing through this encoder block, we obtain a general text feature, $T_{\text{gen}}$, as well as four region-specific text features, $\mathbf{f}_{\text{reg}} = \{f_{\text{reg}}^{\text{L}}, f_{\text{reg}}^{\text{ML}}, f_{\text{reg}}^{\text{MR}}, f_{\text{reg}}^{\text{R}}\}$. Experiments in Sec. 4.4.3 demonstrate the effectiveness of this block.

Additionally, we introduce a Radar-Enrichment Block (REB) to enhance the regional text features with corresponding radar point features, as depicted in Figure 4. The radar point features are initially extracted

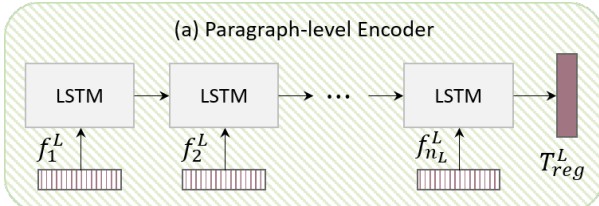

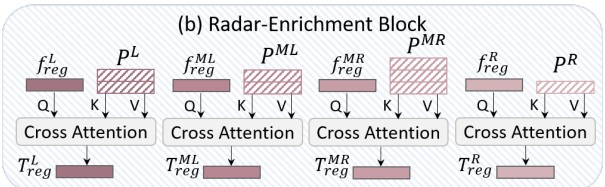

Figure 3: Paragraph-level text encoding block. For a paragraph containing $n$ sentences, the block includes $n$ LSTM layers, which sequentially process the sentences and output a final feature representing the paragraph's global information.

Figure 4: Radar-Enrich block. This block takes regional-specific text features and the radar features as input. The radar point features are divided into four regions to align with the text features. Then, the regional point features are used to enrich the regional text features through cross-attention blocks.

by the PointNet Qi et al. (2017), producing a point feature representation $\mathbf{P} \in \mathbb{R}^{N \times C'_r}$. Based on the spatial positions of radar points, the point cloud can be divided into four regions, corresponding to their 2D projections on the image plane. Thus, the point feature $\mathbf{P}$ can be subdivided into four regional features: $P^L$, $P^{ML}$, $P^{MR}$, and $P^R$. These regional point features enrich the corresponding region-specific text features $f^L_{reg}$, $f^{ML}_{reg}$, $f^{MR}_{reg}$, and $f^R_{reg}$ by cross-attention mechanism Vaswani (2017), resulting in the final region-specific text features $T^L_{reg}$, $T^{ML}_{reg}$, $T^{MR}_{reg}$, and $T^R_{reg}$. Notably, the general text feature $T_{gen}$ remains unaltered throughout this process.

As discussed in Sec. 3.1, the general description includes information about weather conditions. In the nuScenes dataset Caesar et al. (2020), scenes are categorized as normal, rainy, or nighttime. However, during text extraction, certain frames labeled as rainy are sometimes described as "cloudy" in the generated text. This discrepancy limits the weather prediction accuracy for the "rainy" category, which remains around 75% when using only the general text feature, $T_{gen}$. To improve accuracy, we combine the pooled final image feature $f^5_{img}$ from the image encoder with $T_{gen}$, forming a more robust weather feature $T_{wea}$, for the final weather prediction and downstream fusion tasks.

Specifically, $f^5_{img} \in \mathbb{R}^{\frac{H}{32} \times \frac{W}{32} \times C_{img}}$ is first processed by a Global Average Pooling (GAP) layer to reduce the three-dimensional tensor to a vector of length $C_{img}$. Next, an adaptive pooling layer is applied to ensure the image feature matches the dimensions of $T_{gen}$. The pooled image feature is then combined with $T_{gen}$ through element-wise addition, producing the final weather feature, $T_{wea}$. A Multi-layer Perceptron (MLP) is subsequently applied to $T_{wea}$ to predict the final weather classification $\hat{y}$.

### 3.4 Decoder

The decoder takes as input the image features $\{f^i_{img}\}^5_{i=1}$, radar features $\{f^i_{rad}\}^5_{i=1}$, and the weather feature $T_{wea}$, producing the final predicted depth map $\hat{D}$. At each decoding stage, the image, radar, and weather features are fused through our WaFB, illustrated in Figure 5. During the fusion process, $T_{wea}$ provides supplementary weather information to adjust the weighting of the radar branch. Additionally, the text features are fused into the decoding features in two stages, by proposed general attention and the regional attention blocks. Details are as follows:

**Weather-aware Fusion Block:** As depicted in Figure 5, we extend the gated-fusion method Singh et al. (2023) by computing an additional weight informed by weather data. Specifically, $T_{wea}$ is concatenated with the radar feature and passed through a convolutional layer followed by a sigmoid activation to generate the weather-aware weight $\gamma$:

$$\gamma = \sigma(\text{Conv}(T_{wea} \otimes f_{rad})), \tag{1}$$

where $\sigma(\cdot)$ indicates sigmoid activation function and $\otimes$ indicates concatenation. Additionally, the radar feature passes through two separate branches to generate a radar weight $\alpha$, and a projection $\beta$:

$$\alpha = \sigma(\text{Conv}(f_{rad})) \tag{2}$$

$$\beta = \text{ReLU}(\text{Conv}(f_{\text{rad}})). \tag{3}$$

Consequently, the final fused feature is computed as $\alpha \cdot \beta + \gamma \cdot \beta + f_{\text{img}}$.

**Text Features Fusion:** To incorporate $T_{\text{gen}}$ into the fused feature $f_{\text{dec}}^5$ at a scale of $\frac{1}{32}$, we employ a bi-directional cross-attention block (General Attention), as illustrated in Figure 2. The Cross-Attention (CA) block is defined as follows:

$$\text{CA}(Q, K, V) = softmax(\frac{QK^T}{\sqrt{d_k}})V, \tag{4}$$

where $Q, K, V$ indicate query, key, and value, respectively, and $d_k$ is the dimensionality of the keys. In the GA block, $T_{\text{gen}}$ is used as the query while $f_{\text{dec}}^5$ serves as the key and value, generating the first attention feature. Reversely, the second attention feature is obtained by setting $f_{\text{dec}}^5$ as the query and $T_{\text{gen}}$ as the key and value. These two attention features are then concatenated and processed through a convolutional layer to produce a residual feature, which is added back to $f_{\text{dec}}^5$, formulating the output feature $f_{GA}$. The GA block can be formulated as follows:

$$f_{\text{GA}} = \text{Conv}(CA(T_{\text{gen}}W_Q, f_{\text{dec}}^5 W_K, f_{\text{dec}}^5 W_V) \otimes CA(f_{\text{dec}}^5 W_Q, T_{\text{gen}}W_K, T_{\text{gen}}W_V)) + f_{\text{dec}}^5. \tag{5}$$

This resulting feature is subsequently processed by an upsampling module, and a Densely connected Atrous Spatial Pyramid Pooling (DASPP) Yang et al. (2018) module, which enhances contextual feature extraction, yielding $f_{\text{dec}}^4$.

Then, the region-specific text features are integrated into $f_{\text{dec}}^4$. Here, $f_{\text{dec}}^4$ is divided horizontally into four segments (left, middle-left, middle-right, and right), each undergoing bi-directional attention with the corresponding regional text features $T_{\text{reg}}^{\text{L}}$, $T_{\text{reg}}^{\text{ML}}$, $T_{\text{reg}}^{\text{MR}}$, and $T_{\text{reg}}^{\text{R}}$. The resulting four features are concatenated horizontally, forming $(f_{\text{dec}}^4)'$ and passed to the next decoding phase. The UNet-Decoder then takes the decoding feature $(f_{\text{dec}}^4)'$, the intermediate image and radar features $\{f_{\text{img}}^i\}_{i=1}^4$, $\{f_{\text{rad}}^i\}_{i=1}^4$, and the weather feature $T_{\text{wea}}$ as input, estimating the final depth $\hat{D}$.

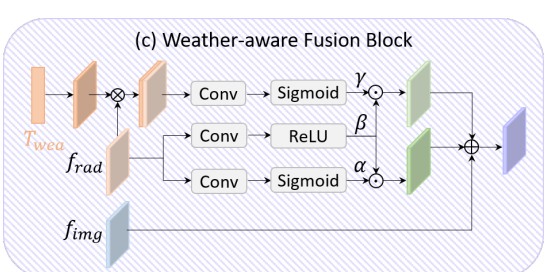

Figure 5: Weather-aware Fusion Block.

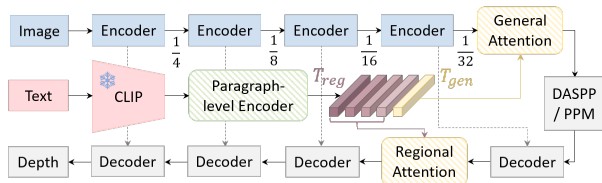

Figure 6: Model structure of integrating text information into image-only methods for KITTI dataset.

### 3.5 Loss Functions

Our model utilizes two loss functions to accomplish the tasks of depth estimation and weather classification. For weather classification, we apply cross-entropy loss to the ground truth label $y$ and the predicted label $\hat{y}$:

$$L_{\text{Cls}} = -\sum_{i=1}^{n} y_i \log \hat{y}_i \tag{6}$$

For depth estimation, we follow the approach in Sun et al. (2025c), using both the dense depth map $D$ and the single-scan depth $D_s$ as supervision:

$$L_{\text{Depth}} = \frac{1}{|\Omega_s|} \sum_{x \in \Omega_s} |D_s(x) - \hat{D}(x)| + \frac{1}{|\Omega|} \sum_{x \in \Omega} |D(x) - \hat{D}(x)|, \tag{7}$$

where the loss is computed only over the set of pixels where $D_s$ or $D$ values are valid.

The final loss is then obtained by summing the two individual losses $L = w_{\text{D}} L_{\text{Depth}} + w_{\text{C}} L_{\text{Cls}}$, where $w_{\text{D}}$ and $w_{\text{C}}$ are the weighting factors.

### 3.6 Text-assisted Monocular Depth Estimation

First, we generate text for the KITTI dataset using our proposed text-generation rules. Without any radar enrichment, the text is processed directly by the frozen pre-trained CLIP text encoder, followed by our proposed paragraph-level encoder, yielding five text features as described in Sec. 3.3. As shown in Figure 6, these features are integrated into the image features through general and regional attention blocks, leading to the final depth prediction.

## 4 Experiments

Table 1: Performance Comparison on the nuScenes Official Test Set.

| Dist* | Method | Sensors | | Model | | Metrics | | | | | | | |
|---|---|---|---|---|---|---|---|---|---|---|---|---|---|
| | | I* | R* | Params↓ | Runtime↓ | MAE ↓ | RMSE ↓ | AbsRel ↓ | log10 ↓ | $RMSE_{log}$ ↓ | $\delta_1$ ↑ | $\delta_2$ ↑ | $\delta_3$ ↑ |
| 50m | RC-PDA Long et al. (2021) | ✓ | ✓ | 4.89M | 0.018s | 2.225 | 4.159 | 0.106 | 0.051 | 0.186 | 0.864 | 0.944 | 0.974 |
| | RC-PDA-HG Long et al. (2021) | ✓ | ✓ | 2.59M | 0.009s | 2.210 | 4.234 | 0.121 | 0.052 | 0.194 | 0.850 | 0.942 | 0.975 |
| | DORN Lo & Vandewalle (2021) | ✓ | ✓ | 107.88M | 0.020s | 1.898 | 3.928 | 0.100 | 0.050 | 0.164 | 0.905 | 0.962 | 0.982 |
| | RadarNet Singh et al. (2023) | ✓ | ✓ | 22.80M | 0.378s | 1.706 | 3.742 | 0.103 | 0.041 | 0.170 | 0.903 | 0.965 | 0.983 |
| | CaFNet Sun et al. (2024b) | ✓ | ✓ | 62.25M | 0.132s | 1.674 | 3.674 | 0.098 | 0.038 | 0.164 | 0.906 | 0.963 | 0.983 |
| | Li *et al.* Li et al. (2024b) | ✓ | ✓ | 36.87M | 0.137s | 1.524 | 3.567 | - | - | - | - | - | - |
| | RadarCam-Depth† Li et al. (2024a) | ✓ | ✓ | - | - | 1.286 | 2.964 | - | - | - | - | - | - |
| | GET-UP Sun et al. (2025c) | ✓ | ✓ | 50.76M | 0.313s | 1.241 | 2.857 | 0.072 | 0.030 | 0.135 | 0.943 | 0.977 | 0.988 |
| | TRIDE (Ours) | ✓ | ✓ | 45.70M | 0.031s | **1.089** | **2.621** | **0.065** | **0.026** | **0.124** | **0.951** | **0.980** | **0.990** |
| 70m | RC-PDA Long et al. (2021) | ✓ | ✓ | 4.89M | 0.018s | 3.338 | 6.653 | 0.122 | 0.060 | 0.225 | 0.822 | 0.923 | 0.965 |
| | RC-PDA-HG Long et al. (2021) | ✓ | ✓ | 2.59M | 0.009s | 3.514 | 7.070 | 0.127 | 0.062 | 0.235 | 0.812 | 0.914 | 0.960 |
| | DORN Lo & Vandewalle (2021) | ✓ | ✓ | 107.88M | 0.020s | 2.170 | 4.532 | 0.105 | 0.055 | 0.170 | 0.896 | 0.960 | 0.980 |
| | RadarNet Singh et al. (2023) | ✓ | ✓ | 22.80M | 0.378s | 2.073 | 4.591 | 0.105 | 0.043 | 0.181 | 0.896 | 0.962 | 0.981 |
| | CaFNet Sun et al. (2024b) | ✓ | ✓ | 62.25M | 0.132s | 2.010 | 4.493 | 0.101 | 0.040 | 0.174 | 0.897 | 0.961 | 0.983 |
| | Li *et al.* Li et al. (2024b) | ✓ | ✓ | 36.87M | 0.137s | 1.823 | 4.304 | - | - | - | - | - | - |
| | RadarCam-Depth† Li et al. (2024a) | ✓ | ✓ | - | - | 1.588 | 3.663 | - | - | - | - | - | - |
| | GET-UP Sun et al. (2025c) | ✓ | ✓ | 50.76M | 0.313s | 1.541 | 3.657 | 0.075 | 0.032 | 0.145 | 0.936 | 0.974 | 0.986 |
| | TRIDE (Ours) | ✓ | ✓ | 45.70M | 0.031s | **1.345** | **3.330** | **0.068** | **0.028** | **0.133** | **0.946** | **0.978** | **0.989** |
| 80m | BTS Lee et al. (2019) | ✓ | | 33.65M | 0.117s | 2.467 | 5.125 | 0.120 | 0.048 | 0.191 | 0.869 | 0.951 | 0.979 |
| | AdaBins Bhat et al. (2021) | ✓ | | 78.45M | 0.284s | 3.541 | 5.885 | 0.197 | 0.089 | 0.261 | 0.642 | 0.929 | 0.977 |
| | P3Depth Patil et al. (2022) | ✓ | | 109.6M | 0.394s | 3.130 | 5.838 | 0.165 | 0.065 | 0.222 | 0.804 | 0.934 | 0.974 |
| | LapDepth Song et al. (2021) | ✓ | | 73.14M | 0.257s | 2.544 | 5.151 | 0.117 | 0.049 | 0.187 | 0.865 | 0.953 | 0.980 |
| | NewCRFs Yuan et al. (2022) | ✓ | | 270.45M | 0.435s | 1.902 | 4.314 | 0.103 | 0.042 | 0.168 | 0.910 | 0.967 | 0.985 |
| | S2D Ma & Karaman (2018) | ✓ | ✓ | 11.49M | 0.007s | 2.374 | 5.628 | 0.115 | - | - | 0.876 | 0.949 | 0.974 |
| | RC-PDA Long et al. (2021) | ✓ | ✓ | 4.89M | 0.018s | 3.721 | 7.632 | 0.126 | 0.063 | 0.238 | 0.813 | 0.914 | 0.960 |
| | RC-PDA-HG Long et al. (2021) | ✓ | ✓ | 2.59M | 0.009s | 3.664 | 7.775 | 0.138 | 0.064 | 0.245 | 0.806 | 0.909 | 0.957 |
| | DORN Lo & Vandewalle (2021) | ✓ | ✓ | 107.88M | 0.020s | 2.432 | 5.304 | 0.107 | 0.056 | 0.177 | 0.890 | 0.960 | 0.981 |
| | RadarNet Singh et al. (2023) | ✓ | ✓ | 22.80M | 0.378s | 2.179 | 4.899 | 0.106 | 0.044 | 0.184 | 0.894 | 0.959 | 0.980 |
| | CaFNet Sun et al. (2024b) | ✓ | ✓ | 62.25M | 0.132s | 2.109 | 4.765 | 0.101 | 0.040 | 0.176 | 0.895 | 0.959 | 0.981 |
| | Li *et al.* Li et al. (2024b) | ✓ | ✓ | 36.87M | 0.137s | 1.927 | 4.610 | - | - | - | - | - | - |
| | RadarCam-Depth† Li et al. (2024a) | ✓ | ✓ | - | - | 1.690 | 3.948 | - | - | - | - | - | - |
| | GET-UP Sun et al. (2025c) | ✓ | ✓ | 50.76M | 0.313s | 1.632 | 3.932 | 0.076 | 0.032 | 0.148 | 0.934 | 0.974 | 0.986 |
| | TRIDE (Ours) | ✓ | ✓ | 45.70M | 0.031s | **1.422** | **3.575** | **0.068** | **0.028** | **0.135** | **0.944** | **0.977** | **0.988** |

\* Dist, I, and R denotes evaluation distance, image, and radar, respectively.
† There is no available code for testing on the nuScenes dataset. Thus, it is not able to calculate the number of parameters and the runtime of the model.

This section starts by explaining the datasets and the details of the implementation. Then, we evaluate the quantitative and qualitative of the depth estimation task. Finally, we conduct several ablation studies to demonstrate the effectiveness of the proposed blocks further.

### 4.1 Datasets and Implementation Details

**nuScenes Dataset.** To validate our TRIDE, we use the nuScenes dataset Caesar et al. (2020), a comprehensive multi-sensor dataset. This dataset includes 700, 150, and 150 scenes in its training, validation, and test subsets, respectively, covering diverse driving conditions such as **normal**, **rainy**, and **night-time** scenarios. For our experiments, we utilize the front camera and single-scan radar as inputs to train and assess the proposed algorithm's effectiveness.

As observed in prior work Li et al. (2024b), accumulating LiDAR frames from past and future instances can introduce significant errors, leading to inaccurate ground truth depth maps. We also found that accumulated dense depth maps struggle to accurately capture the shape of individual objects. Since the dataset lacks official dense ground truth depth maps, developing effective supervision is crucial. In this study, we generate

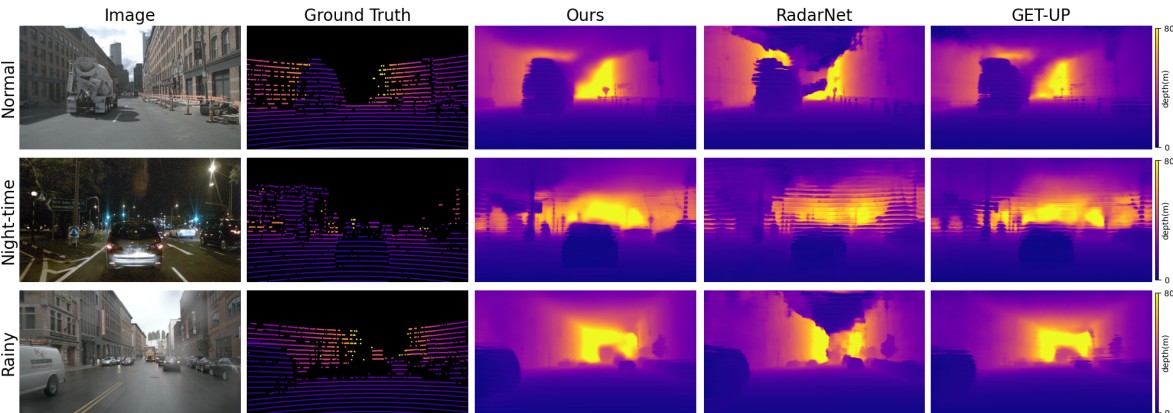

Figure 7: Qualitative comparison on nuScenes test set. Column 1 shows the RGB image; column 2 plots the ground truth depth map. We compared our results with those of the RadarNet and GET-UP at an 80-meter depth range.

a dense depth map $D$ as follows: we first train NewCRFs Yuan et al. (2022) using the single-scan depth map $D_s$, which accurately captures object shapes and produces preliminary dense depth predictions. For each frame, we use NewCRFs to generate $D$ and then refine it by inserting the existing values from $D_s$ into $D$, ensuring more accurate dense depth supervision. Both $D$ and $D_s$ are used to supervise the training of our network. Further details are explained in the supplementary material.

Our model is implemented in PyTorch Paszke et al. (2019) and trained on an Nvidia® Tesla A30 GPU with a batch size of 10. We use the Adam optimizer Kingma & Ba (2017) with an initial learning rate of $1e^{-4}$, adjusting it based on a polynomial decay schedule with a power of $p = 0.9$. To prevent overfitting, we apply data augmentation techniques, including random flips and adjustments to contrast, brightness, and color. Additionally, random cropping to $448 \times 800$ pixels is performed during training to further enhance model robustness.

**KITTI Dataset.** The KITTI dataset Geiger et al. (2012) is the most widely used benchmark for outdoor depth estimation in driving scenarios. We evaluate the efficacy of our proposed text branch by integrating it into existing depth estimation algorithms. The models are retrained and evaluated using the KITTI Eigen split Eigen et al. (2014), with all implementation details consistent with the original publications.

### 4.2 Quantitative Results

**Evaluation on nuScenes.** In this study, we benchmark our proposed TRIDE against existing radar-camera depth estimation methods and camera-only algorithms using standard evaluation metrics. The models are tested on the official nuScenes test set at full image resolution across three distance ranges: up to 50 meters, 70 meters, and 80 meters, as detailed in Table 1. Compared to camera-only methods, incorporating radar data significantly improves depth estimation accuracy while requiring fewer model parameters. Against existing radar-camera algorithms, our TRIDE model achieves superior performance over Sun et al. (2025c), with improvements of 12.25%, 12.72%, and 12.87% in MAE, and 8.26%, 8.94%, and 9.08% in RMSE at evaluation distances of 50, 70, and 80 meters, respectively, underscoring its efficacy in precise depth estimation. Furthermore, we evaluate the computational efficiency of the proposed method: TRIDE comprises 45.70 million parameters, 5 million fewer than the current state-of-the-art, and exhibits a faster runtime, further demonstrating the architectural advantages and efficiency of the proposed approach. Importantly, reported model size and runtime measurements refer solely to the depth estimation stage; text generation and CLIP encoding are evaluated separately. Further details on runtime analysis are provided in the supplementary material.

**Evaluation on KITTI.** We train five monocular depth estimation algorithms by integrating the proposed text branch into the original models, as detailed in Table 2. Standard evaluation metrics, described in the supplementary material, are used for comparison. Integrating the text branch adds approximately 4 million (M) parameters to each model. Nevertheless, in most cases, the models with the added text branch

outperform the originals. For instance, in the case of NewCRFs, a 1.6% increase in parameters results in a 3.1% improvement in RMSE.

Table 2: Quantitative results on the KITTI dataset.

| Model | Abs Rel ↓ | Sq Rel ↓ | RMSE ↓ | $R_{log}$ ↓ | $\delta_1$ ↑ | $\delta_2$ ↑ |
|---|---|---|---|---|---|---|
| BTS Lee et al. (2019) | **0.060** | 0.249 | 2.798 | 0.096 | 0.955 | 0.993 |
| BTS + T | **0.060** | **0.210** | **2.458** | **0.092** | **0.958** | **0.994** |
| NewCRFs Yuan et al. (2022) | 0.052 | 0.155 | 2.129 | 0.079 | 0.974 | 0.997 |
| NewCRFs + T | **0.051** | **0.149** | **2.064** | **0.077** | **0.975** | 0.996 |
| DPT Ranftl et al. (2021) | 0.062 | 0.222 | 2.575 | 0.092 | 0.959 | 0.995 |
| DPT + T | **0.061** | **0.196** | **2.345** | **0.089** | **0.964** | **0.996** |
| PixelFormer Agarwal & Arora (2023) | 0.051 | **0.149** | 2.081 | **0.077** | 0.976 | 0.997 |
| PixelFormer + T | **0.050** | 0.150 | **2.074** | 0.079 | 0.976 | **0.998** |
| IEBins Shao et al. (2024) | **0.050** | **0.142** | 2.011 | 0.075 | 0.978 | 0.998 |
| IEBins + T | **0.050** | 0.143 | **2.006** | **0.074** | 0.978 | 0.998 |

'+ T' indicates integrating text branch inside.

## 4.3 Qualitative Results

Here, we compare the depth maps generated by our TRIDE with those from existing algorithms Singh et al. (2023); Sun et al. (2025c) at a depth range of 80 meters under various weather conditions. It is evident that TRIDE captures more detailed information, such as the shape of the street sign and the concrete mixer in the first row. Additionally, our model demonstrates improved resilience to stripe-wise artifacts at night compared to RadarNet and GET-UP. In rainy conditions, TRIDE accurately captures distant vehicles, including cars and trucks, as shown in the third row.

## 4.4 Ablation Studies

To further demonstrate the efficacy of our proposed methods, we conduct a series of ablation studies to verify the effectiveness of each component.

### 4.4.1 Feature Dimensions Analysis

This section conducts experiments to determine the optimal combination of text feature dimension $C_t$ and radar point feature dimension $C_r'$, while also considering the model's parameter count. The results are presented in Table 3. Note that $C_r' = N/A$ indicates that the paragraph-level text features are extracted without using the radar-enrichment block, further highlighting the efficiency of the proposed block. For optimal effectiveness and efficiency, we select feature dimensions of $C_t = 128$ and $C_r' = 256$.

Table 3: Ablation study on the feature dimension.

| $C_t$ | $C_r'$ | MAE ↓ | RMSE ↓ | AbsRel ↓ | $\delta_1$ ↑ | Params ↓ |
|---|---|---|---|---|---|---|
| 128 | N/A | 1.468 | 3.640 | 0.072 | 0.942 | 44.71M |
| 128 | 64 | 1.430 | 3.582 | 0.070 | 0.944 | 45.45M |
| 128 | 128 | 1.435 | 3.591 | 0.069 | 0.942 | 45.53M |
| **128** | **256** | **1.422** | 3.575 | **0.068** | **0.944** | 45.70M |
| 128 | 512 | 1.432 | 3.589 | 0.069 | 0.943 | 46.03M |
| 256 | N/A | 1.471 | 3.660 | 0.072 | 0.942 | 51.83M |
| 256 | 64 | 1.426 | **3.570** | 0.069 | **0.944** | 53.03M |
| 256 | 128 | 1.437 | 3.571 | 0.070 | 0.943 | 53.18M |
| 256 | 256 | 1.426 | 3.578 | 0.069 | 0.943 | 53.48M |
| 256 | 512 | 1.438 | 3.588 | 0.070 | 0.943 | 54.07M |

$C_t$ is the text branch dimension, $C_r'$ is the radar branch dimension.

### 4.4.2 Modality Analysis

Here, we first analyze the impact of using different modalities on the depth estimation task, selecting the optimal feature dimensions for the text and radar point features based on the findings from the previous section. The results are presented in Table 4. Our TRIDE model achieves a 30.09% improvement in MAE over the image-only model. Furthermore, incorporating the text branch enhances the image-radar model by an additional 8.20% in MAE.

Subsequently, we examine the most effective fusion stages for integrating text and decoding features, specifically determining the optimal placement of the General Attention (GA) and Regional Attention (RA) blocks, presented in Table 6.

Table 4: Ablation study on the modality analysis.

| Modalities | MAE ↓ | RMSE ↓ | AbsRel ↓ | $\delta_1$ ↑ | Params ↓ |
|---|---|---|---|---|---|
| I | 2.034 | 4.553 | 0.103 | 0.905 | 29.7M |
| I + T | 1.872 | 4.213 | 0.086 | 0.918 | 31.7M |
| I + R | 1.549 | 3.767 | 0.073 | 0.936 | 40.4M |
| I + R + T$^-$ | 1.472 | 3.644 | 0.070 | 0.940 | 44.71M |
| I + R + T | **1.422** | **3.575** | **0.068** | **0.944** | 45.7M |

I, R, and T represent image, radar, and text modalities. T$^-$ indicates removing radar point encoder and REB from the text branch.

Table 5: Ablation study on text generation process.

| Text Generation | MAE ↓ | RMSE ↓ | AbsRel ↓ | $\delta_1$ ↑ |
|---|---|---|---|---|
| Captioning | 1.531 | 3.801 | 0.074 | 0.937 |
| Ours | **1.422** | **3.575** | **0.068** | **0.944** |

Table 6: Ablation study on text fusion block placement.

| GA | RA | MAE ↓ | RMSE ↓ | AbsRel ↓ | $\delta_1$ ↑ |
|---|---|---|---|---|---|
| 1/32 | 1/32 | 1.437 | 3.602 | 0.070 | 0.942 |
| **1/32** | **1/16** | **1.422** | **3.575** | **0.068** | **0.944** |
| 1/32 | 1/8 | 1.432 | 3.592 | 0.069 | 0.092 |
| 1/16 | 1/32 | 1.441 | 3.615 | 0.070 | 0.942 |
| 1/16 | 1/16 | 1.430 | 3.596 | 0.069 | 0.943 |
| 1/16 | 1/8 | 1.434 | 3.601 | 0.070 | 0.943 |
| 1/32 | w/o | 1.469 | 3.660 | 0.070 | 0.942 |
| w/o | 1/16 | 1.472 | 3.661 | 0.071 | 0.941 |
| w/o | w/o | 1.476 | 3.632 | 0.071 | 0.942 |

'w/o' indicates removing the certain block from the final model.

### 4.4.3 Text Generation and Encoding Analysis

Here, we first evaluate the effectiveness of our proposed text generation process by comparing it to generating a single image caption using the captioning model from Zeng et al. (2024a), as shown in Table 5. Since only one caption is generated per image, we encode it using the CLIP text encoder and fuse this text feature into the final model via the general attention block, omitting the regional attention block in this process. This paper also proposes a paragraph-level text encoding block to extract a unified feature representation for each paragraph, leveraging the capability of LSTM layers. Thus, we present additional experiments to evaluate the effectiveness of the proposed block, presented in Table 7. Specifically, we compare our approach with alternative methods: averaging the text features within each paragraph (avg), concatenating the text features along the channel dimension followed by average pooling and a convolutional layer (pool), using LSTM layers but averaging all the processed hidden features (LSTM_avg), and more straightforward methods like element-wise addition, multiplication, and attention. The results indicate that simply averaging or applying pooling techniques fails to capture the global key information of the entire paragraph. Additionally, averaging the hidden states of the LSTM disregards both the sequential information and the relative importance of features within the paragraph.

Table 7: Ablation study on the paragraph-level encoding block.

| Text Encoding | MAE ↓ | RMSE ↓ | AbsRel ↓ | $\delta_1$ ↑ |
|---|---|---|---|---|
| avg | 1.544 | 3.838 | 0.074 | 0.936 |
| pool | 1.529 | 3.744 | 0.073 | 0.936 |
| LSTM | **1.422** | **3.575** | **0.068** | **0.944** |
| LSTM_avg | 1.451 | 3.603 | 0.069 | 0.943 |
| addition | 1.469 | 3.651 | 0.070 | 0.941 |
| multiplication | 1.473 | 3.641 | 0.075 | 0.938 |
| attention | 1.437 | 3.579 | 0.070 | **0.944** |

### 4.4.4 Fusion Block Analysis

Table 8 compares the proposed WaFB with concatenation, element-wise addition, and gated fusion Singh et al. (2023).

Table 8: Ablation study on the fusion block.

| Fusion | MAE ↓ | RMSE ↓ | AbsRel ↓ | $\delta_1$ ↑ |
|---|---|---|---|---|
| concatenation | 1.572 | 3.901 | 0.075 | 0.931 |
| addition | 1.601 | 3.929 | 0.075 | 0.928 |
| gated-fusion | 1.526 | 3.737 | 0.072 | 0.936 |
| WaFB | **1.422** | **3.575** | **0.068** | **0.944** |

## 4.5 Quantitative Results under Different Weather Conditions

In this work, we also evaluate the quantitative results under three weather conditions: normal weather, rainy, and night-time. Results under normal weather are presented in the appendix. The test set of nuScenes Caesar et al. (2020) is split into three subsets, containing 4254, 900, and 704 samples, respectively. In the following, we compare our method against RadarNet Singh et al. (2023), GET-UP Sun et al. (2025c), and our model, but with the gated-fusion technique.

(a) Night-time Scenario

| Dist | Method | Metrics | | | |
|------|--------|---------|---|---|---|
| | | MAE ↓ | RMSE ↓ | $\delta_1$ ↑ | $\delta_2$ ↑ |
| 50m | RadarNet Singh et al. (2023) | 2.343 | 4.686 | 0.840 | 0.935 |
| | GET-UP Sun et al. (2025c) | 2.075 | 4.537 | 0.886 | 0.949 |
| | Ours (GF) | 2.069 | 4.443 | 0.883 | 0.953 |
| | Ours | **1.876** | **4.151** | **0.898** | **0.955** |
| 70m | RadarNet Singh et al. (2023) | 2.865 | 5.937 | 0.830 | 0.928 |
| | GET-UP Sun et al. (2025c) | 2.632 | 5.917 | 0.872 | 0.941 |
| | Ours (GF) | 2.582 | 5.631 | 0.869 | 0.947 |
| | Ours | **2.364** | **5.329** | **0.885** | **0.949** |
| 80m | RadarNet Singh et al. (2023) | 3.014 | 6.340 | 0.827 | 0.926 |
| | GET-UP Sun et al. (2025c) | 2.794 | 6.356 | 0.868 | 0.939 |
| | Ours (GF) | 2.725 | 6.001 | 0.866 | 0.945 |
| | Ours | **2.504** | **5.707** | **0.882** | **0.948** |

(b) Rainy Scenario

| Dist | Method | Metrics | | | |
|------|--------|---------|---|---|---|
| | | MAE ↓ | RMSE ↓ | $\delta_1$ ↑ | $\delta_2$ ↑ |
| 50m | RadarNet Singh et al. (2023) | 1.459 | 3.535 | 0.918 | 0.967 |
| | GET-UP Sun et al. (2025c) | 1.018 | 2.605 | 0.954 | 0.979 |
| | Ours (GF) | 0.926 | 2.446 | 0.958 | 0.982 |
| | Ours | **0.891** | **2.421** | **0.960** | **0.982** |
| 70m | RadarNet Singh et al. (2023) | 1.672 | 4.116 | 0.903 | 0.966 |
| | GET-UP Sun et al. (2025c) | 1.204 | 3.162 | 0.950 | 0.977 |
| | Ours (GF) | 1.083 | 2.966 | 0.955 | 0.981 |
| | Ours | **1.042** | **2.926** | **0.958** | **0.981** |
| 80m | RadarNet Singh et al. (2023) | 1.750 | 4.361 | 0.913 | 0.964 |
| | GET-UP Sun et al. (2025c) | 1.275 | 3.392 | 0.949 | 0.977 |
| | Ours (GF) | 1.142 | 3.173 | 0.954 | 0.980 |
| | Ours | **1.102** | **3.138** | **0.957** | **0.981** |

Table 9: Performance comparison under different weather and lighting scenarios.

The results demonstrate that, under rainy conditions, our TRIDE model outperforms the state-of-the-art GET-UP Sun et al. (2025c) by 12.48%, 13.46%, and 13.53% in MAE at evaluation distances of 50, 70, and 80 meters, respectively. Additionally, it is evident that depth estimation algorithms perform the worst under night-time scenarios. However, compared to GET-UP, TRIDE achieves RMSE improvements of 8.51%, 9.94%, and 10.21% at 50, 70, and 80 meters, respectively. Furthermore, replacing gated fusion with our proposed WaFB significantly improves night-time performance, further demonstrating the effectiveness of the proposed fusion method.

Interestingly, when comparing GET-UP and RadarNet Singh et al. (2023), while GET-UP shows significant overall improvements over RadarNet on the full test set, it does not outperform RadarNet in RMSE under night-time scenarios and even performs worse at the 80-meter range. These findings highlight that night-time remains a challenging case for depth estimation. Nevertheless, TRIDE addresses this issue to some extent, demonstrating its robustness in such adverse conditions.

## 5 Conclusion and Future Work

This paper introduces a text generation and feature encoding strategy for depth estimation. To demonstrate the effectiveness of incorporating text information into image-only depth estimation models, we integrate text features into several monocular depth estimation algorithms and train them on the widely used KITTI dataset. The results indicate that adding text information enhances depth estimation accuracy. We further propose TRIDE, a radar-camera depth estimation algorithm trained and evaluated on the comprehensive nuScenes dataset. To leverage the robustness of radar sensors in adverse weather conditions, we introduce a radar-enrichment block that enhances text features with radar point features. Additionally, we propose a weather-aware fusion block that adaptively weights the decoding features from the radar and image branches based on weather conditions, fully utilizing radar's advantages in challenging environments. TRIDE surpasses the previous state-of-the-art model, achieving a 12.87% and 9.08% improvement in MAE and RMSE.

In this paper, we conduct a single MLLM to generate text descriptions by our proposed prompt, demonstrating that incorporating text information improves depth estimation accuracy. In the future, a comparative study across multiple MLLMs will be explored to further assess how the robustness of generated text impacts depth estimation.

## 6 Acknowledgement

Research leading to these results has received funding from the EU ECSEL Joint Undertaking under grant agreement n° 101007326 (project AI4CSM) and from the partner national funding authorities the German Ministry of Education and Research (BMBF).

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

## A  Appendix

The supplementary material first analyzes the text descriptions, including the designed prompts and a comparison between our text descriptions and image captions, in Sec A.1. Next, we examine our generated dense depth maps used for training supervision in Sec A.2. Following this, we detail the evaluation metrics in Sec A.3. Finally, we present a comprehensive comparison of quantitative and qualitative results under different weather conditions in Sec A.4 and Sec A.6, and conclude in Sec A.10.

### A.1  Text Description

#### A.1.1  Prompt Design

In this work, we design a tailored prompt to generate detailed descriptions of a given image. These language-based descriptions serve as prior knowledge of the current scenario, aiding in the understanding of the image. The prompt is formulated as follows:

> **The Designed Prompt.**
> Describe the frame in five parts, and each part starts with a dash sign (-). The first part describes the image in general, including the weather conditions.
> The second to fifth parts describe the objects and estimate their depth (maximum 80 meters) in the right part, middle right part, middle left part, and left part of the image, respectively.

#### A.1.2  Visulization

In this section, we present a visualization of the text description for a given image. As shown in Figure 8, the description consists of five paragraphs. The first paragraph provides an overview of the image, while the second through fifth paragraphs describe the objects present and provide a rough estimate of their depth, progressing from the left to the right of the image. According to the text description, it is easy to align the regional text feature with the image features of the specific regions.

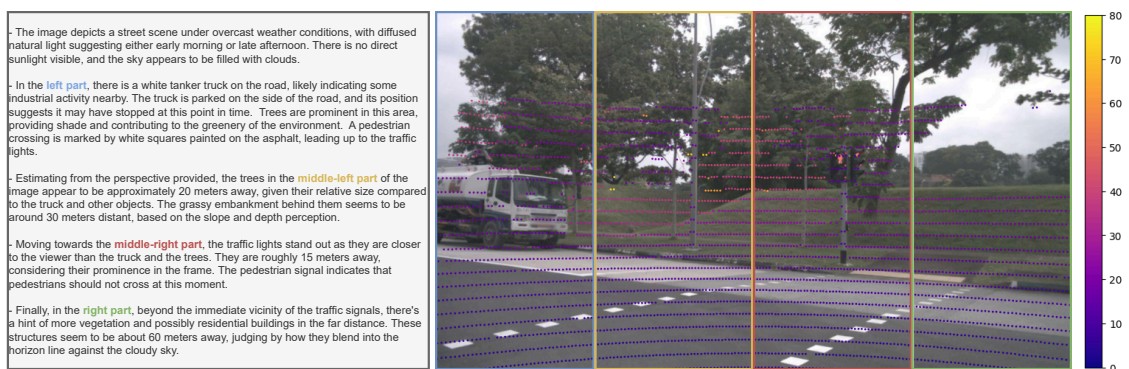

Figure 8: Test description for a given image. The first paragraph describes the image in a general way. Afterwards, the second to the fifth paragraphs introduce the image from left to right.

#### A.1.3  Text Description Comparison

In this work, we also generate image captions using ExpansionNet v2 Hu et al. (2022), which is used in Zeng et al. (2024a;b), for the given image and compare the generated descriptions under three different weather conditions.

**Normal Weather Scenario.**  For the image presented in Figure 8, the generated caption is "*A truck driving down a street with a traffic light*". The descriptions generated using our proposed prompt are as follows:

**Our Text Descriptions.**

- The image depicts a street scene under overcast weather conditions, with diffused natural light suggesting either early morning or late afternoon. There is no direct sunlight visible, and the sky appears to be filled with clouds.

- In the *left* part, there is a white tanker truck on the road, likely indicating some industrial activity nearby. The truck is parked on the side of the road, and its position suggests it may have stopped at this point in time. Trees are prominent in this area, providing shade and contributing to the greenery of the environment. A pedestrian crossing is marked by white squares painted on the asphalt, leading up to the traffic lights.

- Estimating from the perspective provided, the trees in the *middle-left* part of the image appear to be approximately 20 meters away, given their relative size compared to the truck and other objects. The grassy embankment behind them seems to be around 30 meters distant, based on the slope and depth perception.

- Moving towards the *middle-right* part, the traffic lights stand out as they are closer to the viewer than the truck and the trees. They are roughly 15 meters away, considering their prominence in the frame. The pedestrian signal indicates that pedestrians should not cross at this moment.

- Finally, in the *right* part, beyond the immediate vicinity of the traffic signals, there's a hint of more vegetation and possibly residential buildings in the far distance. These structures seem to be about 60 meters away, judging by how they blend into the horizon line against the cloudy sky.

In this simple scenario, the caption provides a general summary of the image. However, our language-based descriptions offer more detailed and contextually relevant information compared to the caption, enriching the understanding of the scene.

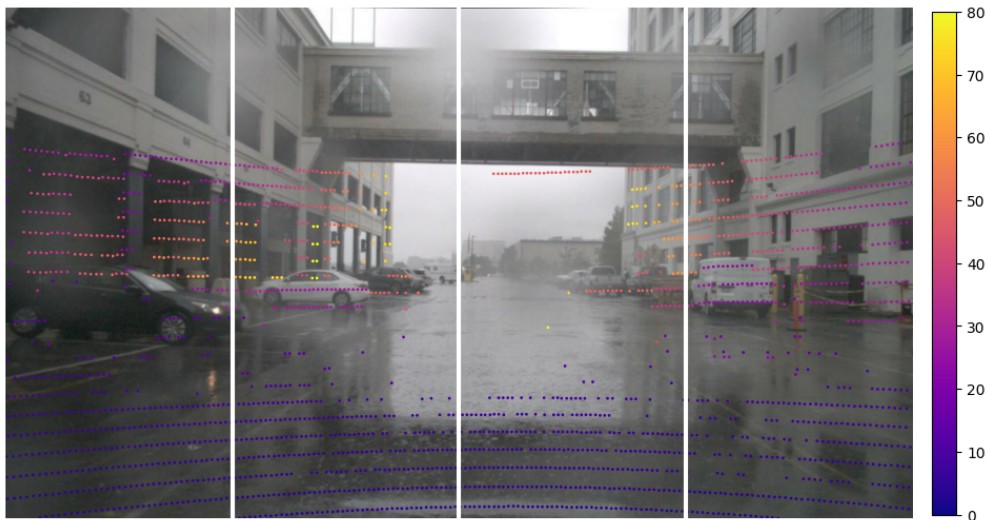

Figure 9: Example of rainy scenario.

**Rainy Scenario.** Figure 9 illustrates a rainy scenario. The generated caption is:

**Image Caption.**
A group of cars parked in the rain under a bridge.

And our generated detailed descriptions are as follows:

**Our Text Descriptions.**

- The image depicts an urban street scene on a *rainy* day. Visibility is somewhat reduced due to the weather, and the wet road reflects the surrounding buildings and vehicles, adding a sense of depth to the scene. The overcast sky suggests that it might be early morning or late afternoon.

- In the *left* part, there are several cars parked along the side of the street, with varying distances from the viewpoint. A dark-colored minivan is closest to the camera, while further away, white and silver sedans can be seen parked in a line. The depth estimation for these objects ranges from approximately 10 meters (minivan) to about 50 meters (furthest sedan).

- In the *middle left* part, more details become visible as we move further down the street. There's a pedestrian walkway overhead connecting two buildings, which adds architectural interest and complexity to the scene. The cars continue to recede into the distance, with their depths estimated at around 30 to 40 meters from the viewer.

- In the *middle right* part, the focus shifts slightly to show additional elements like a white delivery van and some construction barriers indicating ongoing work. These objects appear closer, with the van being roughly 20 to 30 meters away from the viewpoint, and the barriers appearing even nearer, possibly within 10 to 20 meters.

- Finally, in the *right* part, the scene continues with the delivery van and construction activity, suggesting a bustling area despite the inclement weather. Other parked vehicles and the edge of the street are visible, with the van likely being the closest object at about 10 to 15 meters from the camera, and other elements fading into the background at varying distances up to 80 meters.

In more complex scenarios, the captioning model provides only a high-level, generalized description of the image, which is often insufficient for accurate depth estimation. By contrast, our language descriptions offer more detailed and useful information, though they are not without limitations. While the MLLM can effectively identify objects in the image, the prior depth values it predicts often contain significant errors. This underscores the critical role of our radar-enrichment block in correcting and refining depth predictions.

**Night-time Scenario.** Figure 10 illustrates a complex night-time scenario. We compare our text description with the image caption of the given image. From the descriptions, it is evident that the image caption is entirely inaccurate, as there is no train on the road. The captioning model struggles to handle such complex scenes, and incorporating this incorrect information into the depth estimation model could lead to confusion and degraded performance. In contrast, our text descriptions accurately identify that the bus is moving forward and a barrier arm is present on the right, providing reliable and contextually correct information.

**Image Caption.**
A road with a train on the side of it.

And our generated detailed descriptions are as follows:

**Our Text Descriptions.**

- The image depicts an urban setting during what appears to be twilight, with the sky displaying hues of blue and a hint of pink near the horizon. *Street lamps are on*, providing illumination as natural light fades. A bus is parked at what seems like a toll plaza or checkpoint.

- In the *left* part, there's a railing along the edge of the road, leading up to a higher level. The railing suggests that this area might be elevated or a bridge crossing. The street lamp in the middle-left part stands tall, casting its light over the scene. Its position indicates it could be approximately 10 meters from the viewpoint.

- In the *middle left* part, beyond the railing, we can see a gentle slope descending towards the road. This slope likely leads down to another section of the roadway or parking lot. There's a clear delineation between the two levels, suggesting some depth of around 20 meters.

- The *middle right* part features the orange bus prominently. It occupies about half of the frame, indicating its size relative to the surroundings. The bus is positioned such that if it were moving forward, it would travel straight ahead into the distance. Judging by its placement and perspective, it appears to be roughly 30 meters away from the viewpoint.

- In the *right* part, besides the bus, there's a barrier arm partially extended, possibly controlling access to the area. Next to it is a yellow box with signage, which looks like a payment kiosk or information booth. These objects suggest a controlled entry point, perhaps for a paid parking area or a restricted zone. Both the barrier arm and the box seem to be within 40 meters of the viewpoint.

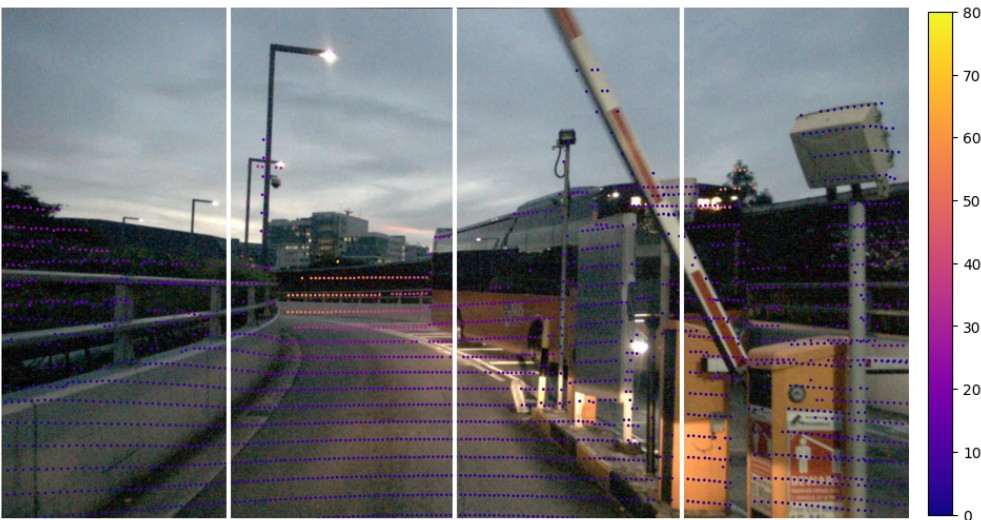

Figure 10: Example of night-time scenario.

## A.2    nuScenes Training Ground Truth Generation

In this section, we compare different depth maps used for supervision. As shown in Figure 11, we evaluate the single-scan depth map, the interpolated dense depth map generated using the method from Singh et al. (2023), and our dense depth map produced by training NewCRFs Yuan et al. (2022).

The NewCRFs approach leverages traditional computer vision techniques, incorporating conditional random field blocks to exploit observation cues such as texture and positional information. This allows it to more effectively detect object contours. From the figure, it is evident that the interpolated depth map is quite noisy compared to the NewCRFs depth maps, with poorly defined object contours. These issues arise from

compensation errors when mapping frames from the past and future to the current frame. However, we also observed that the dense depth maps generated by NewCRFs contain errors. To address this, we integrate the accurate depth values from the single-scan depth map into the NewCRFs-predicted depth map, resulting in a refined dense depth map with reduced noise and improved accuracy.

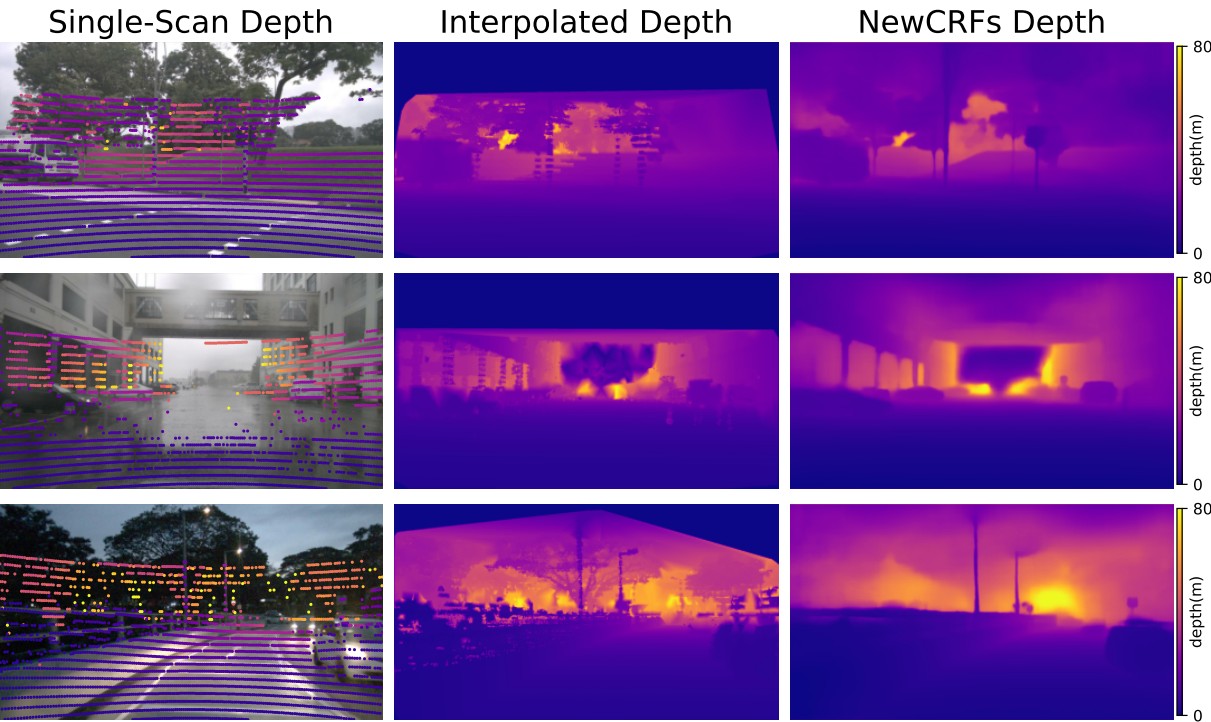

Figure 11: Depth maps comparison. The first column visualizes the single-scan depth map generated by the LiDAR point cloud. The second column visualizes the interpolated depth maps utilizing the method from Singh et al. (2023), and the last column visualizes the dense depth map we use for the supervision.

## A.3 Evaluation Metrics

Table 10 outlines the evaluation metrics employed in this study for comparing performance. Here, $\Omega$ denotes the set of 2D pixels for which ground truth LiDAR depth values are available.

Table 10: Metrics definition for depth estimation task.

| | Definition |
|---|---|
| MAE | $\frac{1}{|\Omega|} \sum_{x \in \Omega} |\hat{d}(x) - d_{gt}(x)|$ |
| RMSE | $(\frac{1}{|\Omega|} \sum_{x \in \Omega} |\hat{d}(x) - d_{gt}(x)|^2)^{1/2}$ |
| AbsRel | $\frac{1}{|\Omega|} \sum_{x \in \Omega} |\hat{d}(x) - d_{gt}(x)|/d_{gt}(x)$ |
| log10 | $\frac{1}{|\Omega|} \sum_{x \in \Omega} |\log_{10} \hat{d}(x) - \log_{10} d_{gt}(x)|$ |
| RMSElog | $\sqrt{\frac{1}{|\Omega|} \sum_{x \in \Omega} ||\log_{10} \hat{d}(x) - \log_{10} d_{gt}(x)||^2}$ |
| $\delta_n$ Thre | $\delta_n = |\{\hat{d}(x) : max(\frac{\hat{d}(x)}{d_{gt}(x)}, \frac{d_{gt}(x)}{\hat{d}(x)}) < 1.25^n\}|/|\Omega|$ |

## A.4 Quantitative Results under Different Weather Conditions

In this section, provide the comprehensive results under different weather conditions, presented in Table 11, Table 12, and 13.

Table 11: Performance Comparison on sub Test Set under Normal Weather.

| Eval Distance | Method | Metrics | | | | | |
|---|---|---|---|---|---|---|---|
| | | MAE ↓ | RMSE ↓ | AbsRel ↓ | RMSElog ↓ | $\delta_1$ ↑ | $\delta_2$ ↑ |
| 50m | RadarNet Singh et al. (2023) | 1.653 | 3.630 | 0.096 | 0.159 | 0.910 | 0.970 |
| | GET-UP Sun et al. (2025c) | 1.150 | 2.632 | 0.065 | 0.122 | 0.950 | 0.982 |
| | Ours (GF) | 1.078 | 2.540 | 0.061 | 0.116 | 0.952 | 0.983 |
| | Ours | **1.001** | **2.410** | **0.059** | **0.113** | **0.958** | **0.984** |
| 70m | RadarNet Singh et al. (2023) | 1.978 | 4.411 | 0.098 | 0.169 | 0.903 | 0.966 |
| | GET-UP Sun et al. (2025c) | 1.432 | 3.387 | 0.068 | 0.131 | 0.944 | 0.979 |
| | Ours (GF) | 1.337 | 3.255 | 0.064 | 0.125 | 0.947 | 0.981 |
| | Ours | **1.240** | **3.084** | **0.061** | **0.121** | **0.953** | **0.982** |
| 80m | RadarNet Singh et al. (2023) | 2.075 | 4.690 | 0.099 | 0.172 | 0.902 | 0.965 |
| | GET-UP Sun et al. (2025c) | 1.516 | 3.645 | 0.069 | 0.134 | 0.942 | 0.979 |
| | Ours (GF) | 1.410 | 3.493 | 0.064 | 0.127 | 0.946 | 0.980 |
| | Ours | **1.311** | **3.314** | **0.062** | **0.123** | **0.952** | **0.981** |

'Ours (GF)' indicates our proposed model with gated-fusion.

Table 12: Performance Comparison on sub Test Set under Rainy Scenario.

| Eval Distance | Method | Metrics | | | | | |
|---|---|---|---|---|---|---|---|
| | | MAE ↓ | RMSE ↓ | AbsRel ↓ | RMSElog ↓ | $\delta_1$ ↑ | $\delta_2$ ↑ |
| 50m | RadarNet Singh et al. (2023) | 1.459 | 3.535 | 0.099 | 0.175 | 0.918 | 0.967 |
| | GET-UP Sun et al. (2025c) | 1.018 | 2.605 | 0.068 | 0.139 | 0.954 | 0.979 |
| | Ours (GF) | 0.926 | 2.446 | 0.063 | 0.130 | 0.958 | 0.982 |
| | Ours | **0.891** | **2.421** | **0.061** | **0.128** | **0.960** | **0.982** |
| 70m | RadarNet Singh et al. (2023) | 1.672 | 4.116 | 0.100 | 0.169 | 0.903 | 0.966 |
| | GET-UP Sun et al. (2025c) | 1.204 | 3.162 | 0.070 | 0.145 | 0.950 | 0.977 |
| | Ours (GF) | 1.083 | 2.966 | 0.064 | 0.136 | 0.955 | 0.981 |
| | Ours | **1.042** | **2.926** | **0.062** | **0.135** | **0.958** | **0.981** |
| 80m | RadarNet Singh et al. (2023) | 1.750 | 4.361 | 0.100 | 0.185 | 0.913 | 0.964 |
| | GET-UP Sun et al. (2025c) | 1.275 | 3.392 | 0.071 | 0.148 | 0.949 | 0.977 |
| | Ours (GF) | 1.142 | 3.173 | 0.064 | 0.139 | 0.954 | 0.980 |
| | Ours | **1.102** | **3.138** | **0.063** | **0.138** | **0.957** | **0.981** |

The results demonstrate that, under rainy conditions, our TRIDE model outperforms the state-of-the-art GET-UP Sun et al. (2025c) by 12.48%, 13.46%, and 13.53% in MAE at evaluation distances of 50, 70, and 80 meters, respectively. Additionally, it is evident that depth estimation algorithms perform the worst under night-time scenarios. However, compared to GET-UP, TRIDE achieves RMSE improvements of 8.51%, 9.94%, and 10.21% at 50, 70, and 80 meters, respectively. Furthermore, replacing gated fusion with our proposed WaFB significantly improves night-time performance, further demonstrating the effectiveness of the proposed fusion method.

Interestingly, when comparing GET-UP and RadarNet Singh et al. (2023), while GET-UP shows significant overall improvements over RadarNet on the full test set, it does not outperform RadarNet in RMSE under night-time scenarios and even performs worse at the 80-meter range. These findings highlight that night-time remains a challenging case for depth estimation. Nevertheless, TRIDE addresses this issue to some extent, demonstrating its robustness in such adverse conditions.

## A.5  Convergence Curve of Loss Functions

In this section, we plot the convergence curves of the losses used to train our TRIDE. As described in Sec. 3.5, we employ two loss terms: a classification loss and a depth estimation loss, the latter being composed of single-scan depth supervision and dense depth supervision. Figure 12 shows the evolution of these losses during training. It is evident that the classification loss converges within a few thousand steps, whereas the depth losses require substantially more iterations to stabilize. Moreover, the depth loss values are markedly larger in magnitude than the classification loss; accordingly, in the final loss function, we set $w_C = 1$ and $w_D = 1$ during training.

Table 13: Performance Comparison on sub Test Set under Night-time Scenario.

| Eval Distance | Method | Metrics | | | | | |
|---|---|---|---|---|---|---|---|
| | | MAE ↓ | RMSE ↓ | AbsRel ↓ | RMSElog ↓ | $\delta_1$ ↑ | $\delta_2$ ↑ |
| 50m | RadarNet Singh et al. (2023) | 2.343 | 4.686 | 0.150 | 0.235 | 0.840 | 0.935 |
| | GET-UP Sun et al. (2025c) | 2.075 | 4.537 | 0.117 | 0.212 | 0.886 | 0.949 |
| | Ours (GF) | 2.069 | 4.443 | 0.118 | 0.199 | 0.883 | 0.953 |
| | Ours | **1.876** | **4.151** | **0.109** | **0.189** | **0.898** | **0.955** |
| 70m | RadarNet Singh et al. (2023) | 2.865 | 5.937 | 0.154 | 0.251 | 0.830 | 0.928 |
| | GET-UP Sun et al. (2025c) | 2.632 | 5.917 | 0.123 | 0.231 | 0.872 | 0.941 |
| | Ours (GF) | 2.582 | 5.631 | 0.124 | 0.213 | 0.869 | 0.947 |
| | Ours | **2.364** | **5.329** | **0.115** | **0.203** | **0.885** | **0.949** |
| 80m | RadarNet Singh et al. (2023) | 3.014 | 6.340 | 0.155 | 0.255 | 0.827 | 0.926 |
| | GET-UP Sun et al. (2025c) | 2.794 | 6.356 | 0.125 | 0.236 | 0.868 | 0.939 |
| | Ours (GF) | 2.725 | 6.001 | 0.125 | 0.217 | 0.866 | 0.945 |
| | Ours | **2.504** | **5.707** | **0.116** | **0.207** | **0.882** | **0.948** |

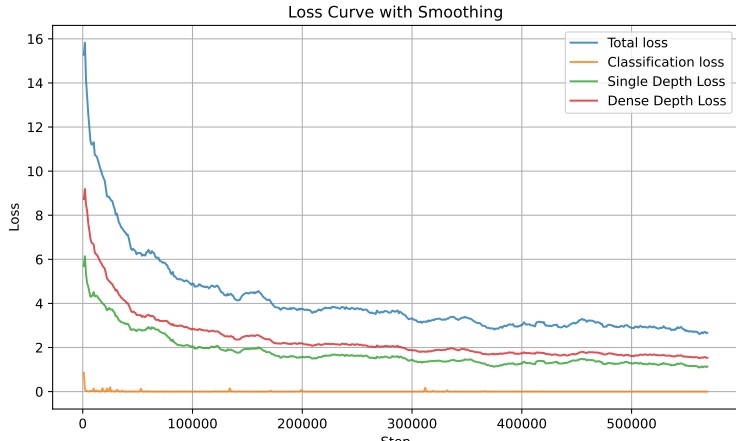

Figure 12: Loss Convergence Curve.

## A.6 Qualitative Results under Different Weather Conditions

In this section, we further analyze the qualitative results under different weather conditions. First, we show results under rainy scenarios. Then we illustrate more results during night-time.

### A.6.1 Rainy Scenarios

As shown in Figure 13, the first column displays the image with the projected single-scan ground truth depth, followed by the predicted depth maps from our TRIDE, RadarNet Singh et al. (2023), and GET-UP Sun et al. (2025c) at a range of 80 meters. From the figures, it is evident that our model predicts more detailed information. For instance, in the first, second, and fourth rows, TRIDE provides more accurate contours of humans. In the third row, our model is the only one capable of correctly predicting the track, while the others fail to distinguish between the upper part of the track and the sky. The last row demonstrates that our model produces more continuous depth maps compared to the others.

### A.6.2 Night-time Scenarios

In this section, we analyze the predicted depth maps under night-time scenarios, as illustrated in Figure 14. As discussed in Sec 4.5, predicting accurate depth maps at night poses significant challenges. Here, we compare the depth maps predicted by our TRIDE model with those from RadarNet and GET-UP.

Overall, our method demonstrates superior performance in detecting object shapes and identifying objects at farther ranges. For instance, in the first and second rows, TRIDE accurately captures the contours of the bus and cars. In the third row, our model is the only one to correctly detect the bus at a far distance. Additionally, TRIDE excels in capturing detailed information, as shown in the fourth row, where it accurately separates the human and the bus. In contrast, the other two methods predict almost the same depth for both,

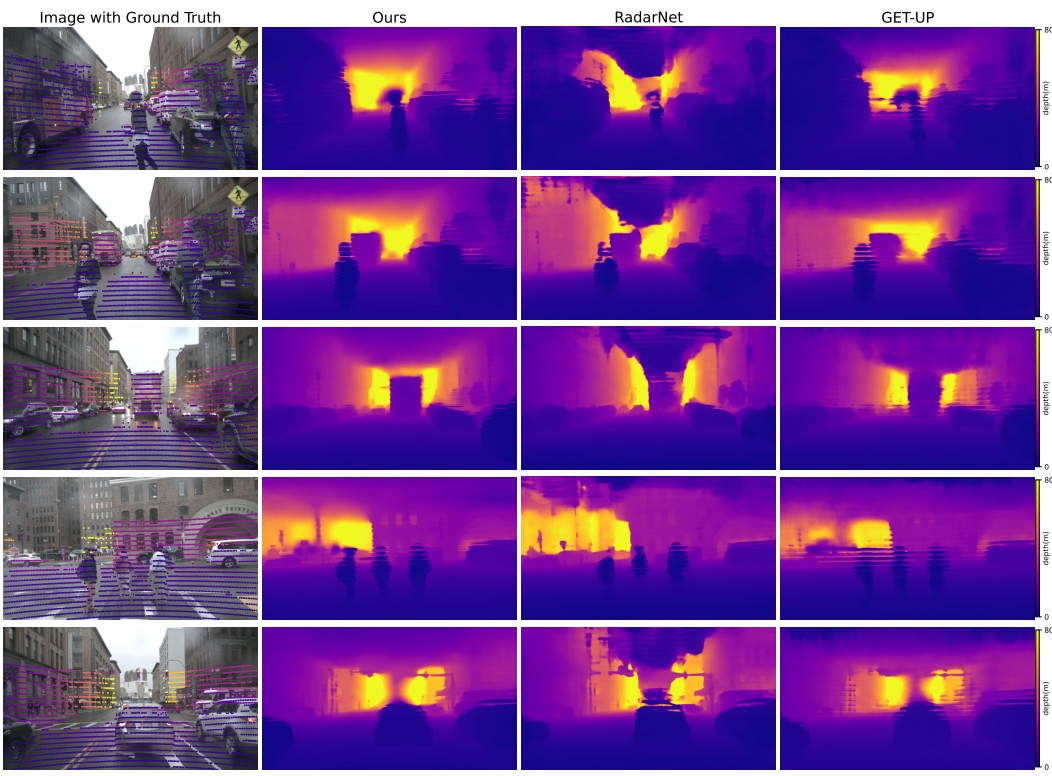

Figure 13: Qualitative comparison under rainy scenarios.

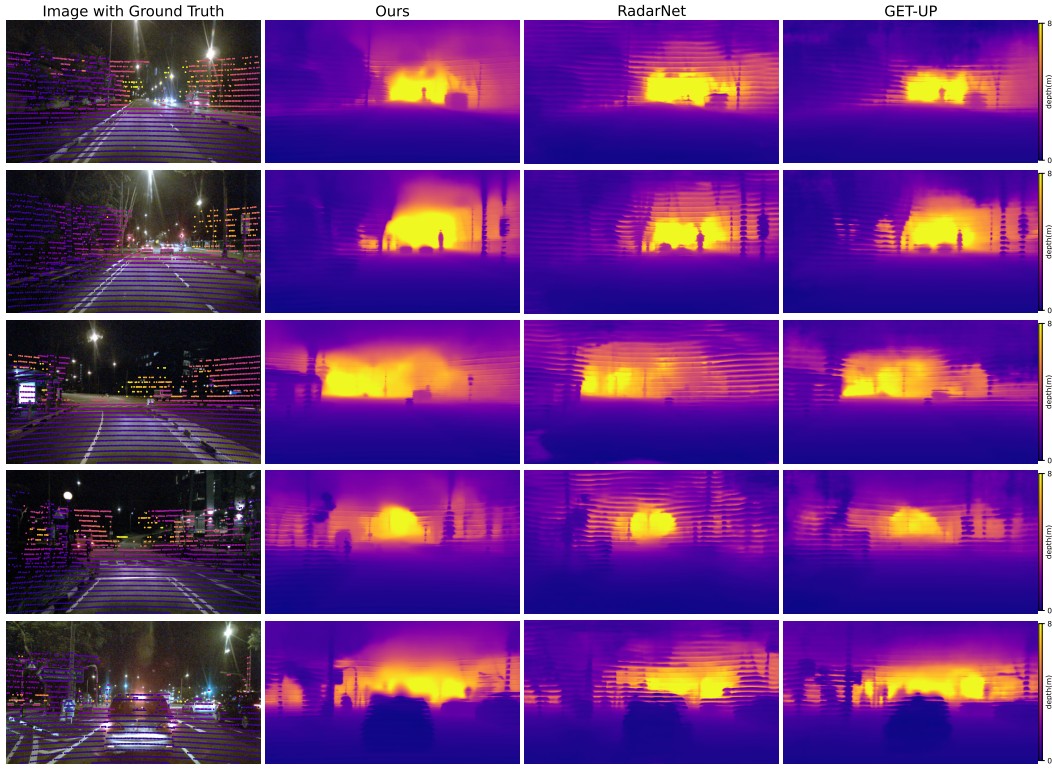

Figure 14: Qualitative comparison under night-time scenarios.

despite the bus being much farther away in the image. Furthermore, our model produces more continuous depth predictions, as demonstrated in the fifth row.

### A.6.3 Normal Weather Scenarios

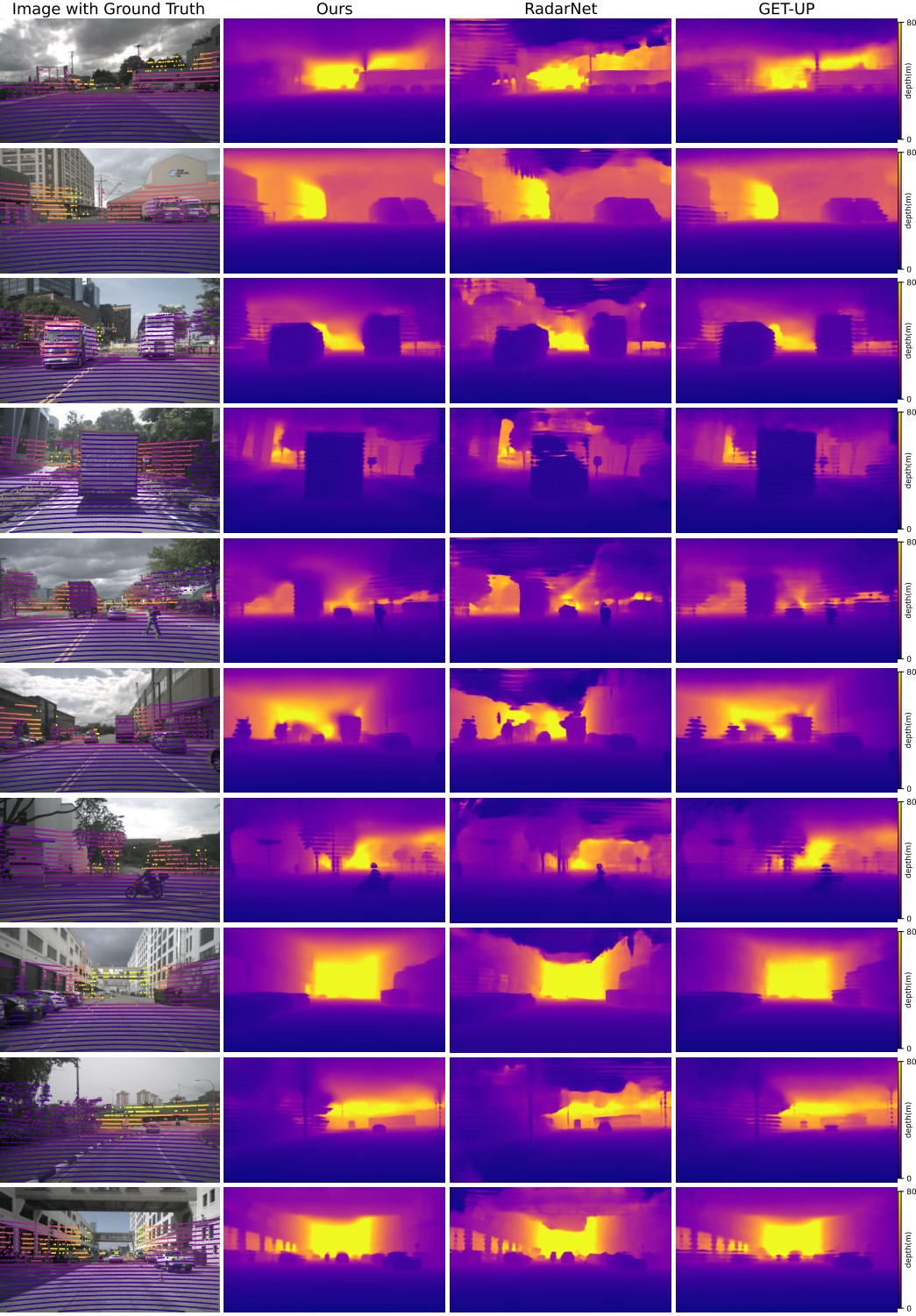

Figure 15: Qualitative comparison under normal weather conditions.

Here, we present additional depth maps under normal weather conditions, as shown in Figure 15. From the figures, it is evident that our method predicts more continuous depth maps and effectively separates nearby objects. For example, in the second row, the upper parts of the two parked trucks are better distinguished compared to the other two methods. Additionally, the depth of the nearby parked cars in the last row is more accurately estimated.

For smaller objects, such as the human in the fifth row and the motorcycle in the seventh row, our method excels in capturing the shape of the objects with greater accuracy. Furthermore, at farther distances, our method continues to predict reasonable shapes and accurate depth values. For instance, in the ninth row, our method successfully recognizes the bus at the farthest range, whereas the other methods fail to do so.

### A.7 Failure Case Analysis

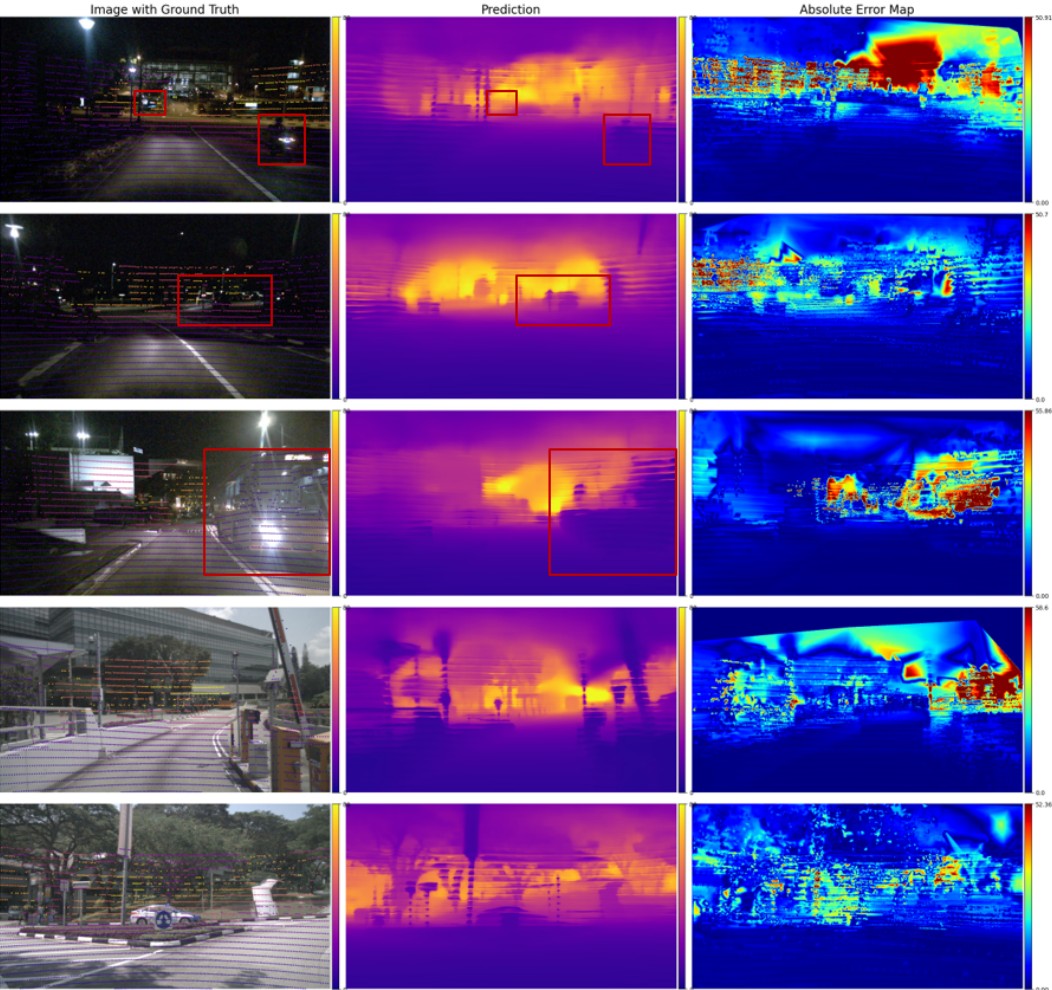

Figure 16: Failure Cases Visualization.

In this section, we examine failure cases of our depth estimation, as illustrated in Figure 16. We also compute the absolute error map between the predicted depth and the interpolated dense depth ground truth. As noted in Sec. 4.5, prediction accuracy degrades at night. In the first three rows of Figure 16, low image quality and challenging lighting prevent accurate shape prediction—e.g., the motorcycle in the first row and the bus in the third row. We also observe that scenes lacking salient objects hinder reliable depth estimation, since the network has no reference points. Conversely, cluttered scenes with many vertical elements (street lamps, road signs, etc.) further complicate predictions. Addressing these issues remains important future work.

## A.8 Generalizability of the Text Branch

We introduce a text branch and integrate it, together with the General Attention and Regional Attention modules, into an existing camera–radar depth estimation framework.

Specifically, we augment CaFNet Sun et al. (2024b), an open-source end-to-end trainable model, while retaining its original training strategy. In contrast, most other methods Singh et al. (2023); Li et al. (2024a); Long et al. (2021) follow a two-stage pipeline, which complicates training, and as noted in Sun et al. (2025c), incurs substantially longer training times. We therefore defer integration with those approaches to future work. As shown in Table 14, incorporating the text branch into CaFNet yields a clear performance improvement.

Table 14: Integrating Text Branch into Existing Model.

| Model | MAE ↓ | RMSE ↓ | AbsRel ↓ | $\delta_1$ ↑ |
|---|---|---|---|---|
| CaFNet Sun et al. (2024b) | 2.109 | 4.765 | 0.101 | 0.895 |
| CaFNet + T | 2.096 | 4.701 | 0.101 | 0.897 |

## A.9 Runtime Analysis

Following WorDepth Zeng et al. (2024a), we generate all text descriptions offline and persist them to disk. Subsequently, these text descriptions are encoded using a frozen CLIP model, and the resulting text features are stored locally. This pipeline ensures that both text generation and encoding are performed only once per dataset, thereby eliminating redundant computation during model training and inference. In our architecture, the depth-estimation network's text branch begins with paragraph-level encoding (see Figure 2), consuming the precomputed CLIP features as input. All timings were measured on a single NVIDIA A30 GPU. Text generation requires 9.4 s per frame, and CLIP encoding requires 0.235 s per frame. Thereafter, the standalone depth-estimation network, including the text branch starting from the paragraph-level encoding, runs in 0.031 s per frame.

## A.10 Conclusion and Future Work

In this work, we present TRIDE, a novel approach that leverages text descriptions to support radar-camera depth estimation, achieving state-of-the-art performance on the nuScenes dataset. As highlighted in the previous sections, night-time poses significant challenges for accurate depth prediction. This underscores the importance of our proposed weather-aware fusion block, as images captured at night are less reliable, while radar remains robust under such conditions.

**Future Work**  From the results in Table 11, Table 12, and Table 13, it is evident that depth estimation performs significantly worse at night-time, with MAE more than doubling compared to rainy scenarios. Improving depth estimation under night-time conditions is therefore a critical area for future research.

Additionally, in this paper, we propose a tailored prompt design and leverage a multimodal large language model for text generation, improving alignment between textual and visual features. Nonetheless, producing consistently informative descriptions for diverse perception tasks remains challenging. Future work may investigate the impact of such descriptions on other tasks (e.g., object detection, semantic segmentation) and develop methods to generate more task-relevant text. Moreover, reducing the latency of reliable text generation will be essential to support real-time applications.

