# OpenReview forum: "TRIDE: A Text-assisted Radar-Image weather-aware fusion network for Depth Estimation"
_TMLR — Accepted by TMLR_

### Review · Reviewer_TpFQ · 2025-06-10

**Summary Of Contributions:**

The paper proposes TRIDE, a novel multi-modal framework that integrates radar, image, and text information to enhance depth estimation, particularly under adverse weather conditions. A tailored text generation strategy produces both global and region-specific descriptions, which are encoded and fused into the model using attention mechanisms. To improve robustness, the authors introduce a radar-enrichment block that refines text features with radar data, and a weather-aware fusion block that adaptively balances sensor inputs based on environmental conditions. TRIDE achieves state-of-the-art results on the nuScenes dataset, demonstrating notable improvements in both accuracy and efficiency over existing methods.

**Audience:**

Yes

**Broader Impact Concerns:**

There is no broader impact concerns for this paper.

**Claims And Evidence:**

Yes

**Requested Changes:**

1.The use of LSTM for paragraph-level feature fusion is interesting, though it may benefit from additional justification. It would be helpful to include comparisons with simpler alternatives such as addition, concatenation, or multiplication, as well as with transformer-based encoding. These comparisons would strengthen confidence in the current design choice.

2.The Radar-Enrich Block is a key component of your model. Including some form of quantitative measurement or visual evidence of the text-radar alignment would help validate its contribution and provide a clearer understanding of the fusion effectiveness.

It would be valuable to investigate whether the proposed approach maintains its effectiveness when used with different radar backbones. For instance, testing with architectures like VoxelNet [1] could demonstrate the generality of your design.

3.The current description in Section 3.4 may be difficult to follow for some readers. Adding more detailed mathematical formulations could help improve clarity and ensure a more precise understanding of the proposed mechanism.

4.Including convergence curves for the two loss functions used in training would provide helpful insight into the model’s optimization behavior and training stability.

5.Please double-check that all hyperparameters are clearly stated in the implementation details. For example, the weights used to combine the two losses do not appear to be mentioned and should be added for completeness and reproducibility.

6.Since most existing approaches do not rely on textual descriptions, it would be interesting to see how performance changes when simply adding a text branch to existing methods like GET-UP. Such a comparison could help clarify the unique contribution of your fusion strategy beyond just the inclusion of textual data.

7.It would be appreciated if you could clarify whether the runtime figures reported include the time for generating textual descriptions. Providing a more detailed runtime analysis would also help readers better understand the trade-off between performance and computational cost.

8.The results would benefit from deeper analysis explaining why your design leads to strong performance. Connecting the observed improvements more explicitly to the novel aspects of your model would enhance the reader’s understanding of its effectiveness.

9.In the section evaluating KITTI, the table citation appears to be incorrect. Additionally, providing a more thorough discussion of the results in this section would make the evaluation more informative.

10. Including a few failure cases in the qualitative results would allow for a more balanced view of the model’s limitations and could help guide future improvements.

**Strengths And Weaknesses:**

Strengthens:
1. The proposed method achieves promising performances on the two leveraged dataset, demonstrating the superior performance to achieve robust depth estimation by using three modalities. The proposed method could benefit the community.

2. The authors have conducted ablation study to illustrate the effectiveness of each propose module.

3. The paper is generally well-written and easy to understand.

Weaknesses:

1. The authors propose to use LSTM to achieve paragraph level feature fusion, how about direct fusion using addition, concatenation, and multiplication or using transformer block to encoder it? The authors are suggested to provide more ablations regarding this design,

2. Could the authors provide some quantitative measurement regarding the text-radar alignment of their trained model regarding the Radar-Enrich Block? or could it be visualized somehow?

3. Lack of ablation regarding the radar feature extraction backbone, can it work well for other architectures, e.g., VoxelNet?
[1] Zhou, Y., & Tuzel, O. (2018). Voxelnet: End-to-end learning for point cloud based 3d object detection. In Proceedings of the IEEE conference on computer vision and pattern recognition (pp. 4490-4499).

4. The authors are suggested to add more formulas in Sec 3.4 to make the description clearer

5. The authors are recommended to show the convergence trajectories based on these two loss functions.

6. The authors are suggested to double check if all the hyperparameters are mentioned in the implementation details section. The weights of the two losses are not mentioned in the implementation details.

7. Most of the existing approaches do not rely on the textual description generation, I am wondering how about the performances if we simply add text branch on them, e.g., using text branch on GET-UP, could the proposed method outperforms this variant?

8. Regarding the runtime of the proposed method, did the authors take the description generation part into consideration? The authors are also encouraged to add more analysis regarding the runtime.

9. Regarding the quantitative results analysis, the authors are encouraged to conduct anaylsis regarding why their method can achieve such good performances based on the novel designs introduced. More in-depth analysis should be provided.

10. Regarding the evaluation on KITTI section, the table is not correctly citated, the authors are also encouraged to add more analysis.

11. Failure cases are suggested to be provided in the qualitative results to enable limitation analysis.

---

> ### Author Response · Authors · 2025-06-18
> **Response to reviewer TpFQ (part 1)**
>
> Dear Reviewer,
>
> We sincerely thank you for your thoughtful and constructive feedback, which has helped us improve the clarity and rigor of our manuscript. We are glad that you recognize the novelty of our approach and the comprehensiveness of our experimental evaluation. Below, we address each of your points in detail.
>
> ### **Response Point 1:**
>
> Thank you for suggesting further experiments with addition, multiplication, and attention-based fusion. We conducted these experiments and included the results in Table 7. Regarding concatenation, as noted in Section 4.4.3, concatenating frame-wise text features yields variable-length vectors across frames. To mitigate this, we experimented with concatenation followed by average pooling and a convolutional layer (denoted “pool” in Table 7).
>
> ### **Response Point 2:**
> Most radar–camera depth estimation methods first project radar points onto the image plane and then encode the resulting pseudo-image using a 2D backbone (e.g., ResNet). In our case, we additionally employ a lightweight PointNet to extract features directly in 3D, enabling more effective interaction with our text branch. Architectures such as VoxelNet are designed for dense LiDAR point clouds, and encoding the point cloud in the Bird’s eye view space.
> First of all, when we are doing depth estimation task, encoding the point cloud in the BEV space is not needed. Secondly, on nuScenes, radar data lack reliable height measurements and consist of around 100 points per frame, so voxelization yields predominantly empty voxels and degrades performance.
>
> ### **Response Point 3:**
> We agree that Section 3.4 would benefit from greater clarity. In the revised manuscript, we reorganize the presentation, introduce additional notation, and provide formulae to guide the reader through the process.
>
> ### **Response Point 4:**
> Thank you for recommending convergence curves. We include plots of both depth and classification losses in the supplementary material (A.4) to illustrate the training dynamics more transparently.
>
> ### **Response Point 5:**
> As observed in the convergence curves, the classification loss converges quickly and is much smaller than the depth estimation loss. Consequently, we set both weights to one ($w_D = 1, w_C = 1$) to simplify hyperparameter tuning without adversely affecting depth training. We clarify this rationale in the revised manuscript (A.4).
>
> ### **Response Point 6:**
> We thank the reviewer for suggesting the integration of the text branch into existing radar–camera methods. Accordingly, we have incorporated our text branch, together with the General Attention and Regional Attention blocks, into CaFNet [1]. Due to time constraints and the substantially longer training time required by GET-UP [2], we have not yet retrained that model; we leave this for future work. Nevertheless, our experiments confirm that the proposed modules consistently enhance model performance. These additional results are reported in A.7.
>
> ### **Response Point 7:**
> We have conducted a more detailed runtime analysis, which will be included in the revised manuscript (A.8). Following WorDepth [3], we generate all text descriptions offline and persist them to disk. Subsequently, these text descriptions are encoded using a frozen CLIP model, and the resulting text features are likewise stored locally. This pipeline ensures that both text generation and encoding are performed only once per dataset, thereby eliminating redundant computation during model training and inference. In our architecture, the depth‐estimation network’s text branch begins with paragraph‐level encoding (see Figure 2), consuming the precomputed CLIP features as input.
> All timings were measured on a single NVIDIA A30 GPU. Text generation requires 9.4 s per frame, and CLIP encoding requires 0.235 s per frame. Thereafter, the standalone depth‐estimation network, including the text branch starting from the paragraph-level encoding, runs in 0.031 s per frame. We would like to emphasize that generating the text description now still takes a much longer time. However, the field of vision–language modeling is advancing rapidly: a growing number of compact architectures achieve inference times on the order of milliseconds, especially when combined with quantization and pruning. Integrating such lightweight, quantized VLMs into our pipeline is therefore planned as future work to bridge the gap between our prototype and real-world, real-time applications.
>
> [1] Cafnet: A confidence-driven framework for radar camera depth estimation.In 2024 IEEE/RSJ International Conference on Intelligent Robots and Systems (IROS)
>
> [2] Get-up: Geometric- aware depth estimation with radar points upsampling.In 2025 IEEE/CVF Winter Conference on Appli- cations of Computer Vision (WACV)
>
> [3] Wordepth: Variational language prior for monocular depth estimation. In Proceedings of the IEEE/CVF Conference on Computer Vision and Pattern Recognition

---

> ### Author Response · Authors · 2025-06-18
> **Response to reviewer TpFQ (part 2)**
>
> ### **Response Point 8:**
>
> We organize our main contributions and their empirical support as follows. To demonstrate the efficacy of each component, we perform extensive ablation studies alongside both quantitative and qualitative evaluations.
> 1. Novel Text Branch
> - We propose a text branch that combines prompt generation with CLIP-based feature extraction, augmented by two attention mechanisms—General Attention (GA) and Regional Attention (RA)—to infuse language cues into image features.
> - Integration into camera-only depth estimators on KITTI yields the results reported in Table 2.
> - Table 5 compares our prompt-generation strategy against the WorDepth [3] image-captioning baseline, and Table 6 identifies the optimal insertion points for GA and RA.
> - An ablation study in Table 7 shows that our encoding pipeline outperforms alternative text-encoding methods.
> - In the Appendix, we detail our prompt‐design methodology and visualize sample prompts alongside single-image captions.
> 2. TRIDE: Multi-Modal Radar–Text–Image Framework
> - We introduce TRIDE, a unified radar–text–image architecture that achieves state-of-the-art results on nuScenes.
> - Quantitative metrics (Table 1) and qualitative examples (Figure 7) confirm its superior performance, and—per reviewer suggestion—we include failure-case analysis in the revised manuscript (A.6). Additionally, more qualitative comparisons under different weather conditions are visualized in A.5.
> - In Table 4, we further dissect the individual contributions of radar, camera, and text modalities, demonstrating that the text branch yields significant gains with minimal parameter overhead.
> 3. Robust Fusion Modules
> - To exploit radar’s resilience, we design the Radar-Enriched Block (REB), which enriches regional text features with raw radar point features.
> - We also develop the Weather-Aware Fusion Block (WaFB), which conditions feature fusion on prevailing weather.
> - Table 4 shows that adding REB improves depth‐estimation accuracy without a substantial increase in parameters, and Table 8 demonstrates that WaFB outperforms existing fusion schemes.
> - Finally, Tables 9 in the revised manuscript validate our framework’s robustness under diverse adverse-weather scenarios, and accompanying qualitative results (segmented by weather) further corroborate these advantages.
>
>
> ### **Response Point 9:**
> Thanks for pointing that out. We are sorry for making this mistake. We have already fixed it, and the table is added to the revised manuscript (Table 2).
>
> ### **Response Point 10:**
> We agree that failure-case examples will provide valuable insights. Accordingly, we have included a new appendix section (A.6) illustrating common errors under challenging conditions, along with discussions of their causes.
>
>
> Once again, thank you for your thorough review and helpful suggestions. We believe that these revisions significantly strengthen our work and look forward to your further feedback.

---

### Review · Reviewer_yVLu · 2025-06-13

**Summary Of Contributions:**

This paper proposes TRIDE, a new depth estimation network that fuses three modalities: images, radar, and text. Inspired by recent radar and image fusion research, this study focuses on the fusion of three modalities, whereas previous studies have been limited to two modalities, and multi-modal learning that considers text has not been proposed.The contributions of this study lie in its focus on the fusion of these three modalities, particularly the use of text encoders (sentence-level features), the proposal of the Radar-Enrichment Block (REB) and the Weather-aware Fusion Block (WaFB), and the effective integration of these components to maintain consistency. The effectiveness of the proposed method is evaluated using standard radar point cloud datasets such as nuScenes.

**Audience:**

Yes

**Broader Impact Concerns:**

It may be worth mentioning that MLLM occasionally misdetects, which can cause significant deviation in depth estimation and increase the risk of collision. I recommends explicitly stating that the introduction of a fail-safe mechanism and uncertainty estimation is essential.

**Claims And Evidence:**

Yes

**Requested Changes:**

1. I recommends including Table 10-12 in the main text.
2. Although Fig. 1 is useful for understanding the overall purpose, it is not referenced anywhere. It would be helpful to clarify which blocks in Fig. 1 correspond to the proposed REB and WaFB blocks mentioned in the introduction. Please cite it correctly in the introduction. In addition, there is inconsistency in terminology between Fig. 1 and Fig. 2. For example, the terms 'Radar Points Projection' and 'Text Encoder' in Fig. 1 are not included in Fig. 2. 'Depth Estimation Decoder' is simply referred to as 'Decoder' in Fig. 2.
3. Table ?? is included on page 9.
4. I strongly recommends that the source code be made publicly available. This will enhance the credibility of the paper

**Strengths And Weaknesses:**

Strengths:
- This is the first study to utilize text information in radar fusion, and there are few studies focusing on weather adaptation, making this research inherently beneficial for practical applications.
- The authors proposed unique blocks such as REB and WaFB and demonstrated their effectiveness.
- The evaluation is comprehensive.
- The evaluations are presented in Tables 10-12 for normal, rainy, and nighttime scenarios, demonstrating consistent effectiveness in each scenario. Furthermore, the fact that the nighttime scenario is more difficult than the other scenarios provides a useful perspective for future research. This would simply strengthen the work in my view.

Weaknesses:
- Please refer to 2 and 3 in Requested Changes below. Although these are minor, they could undermine the reliability of the paper. The authors should check the content thoroughly.

---

> ### Author Response · Authors · 2025-06-18
> **Response to Reviewer yVLu**
>
> We thank Reviewer yVLu for their positive evaluation and for recognizing both the novelty of our approach and the comprehensiveness of our evaluation. Below, we address each of your comments in turn.
>
> ### **Point 1: Relocation of Tables 10–12 into the Main Text:**
> We appreciate this suggestion. In the revised manuscript, we move the evaluations currently presented in Tables 10–12 from the appendix into the main text (Table 9), thereby more directly illustrating the effectiveness of our algorithm under diverse weather conditions.
>
> ### **Point 2: Clarification and Alignment of Figure 1:**
> Thank you for noting that Figure 1 was not properly introduced in the Introduction and that its caption and content are misaligned. We revise the manuscript to ensure that Figure 1 is clearly referenced upon first mention, that its caption accurately reflects its content, and that the figure and caption are fully synchronized.
>
> ### **Point 3: Correction of Table Reference on Page 9:**
> We apologize for the oversight on page 9. The current placeholder should refer to “Table 2,” which reports the quantitative results of integrating the text branch into camera-only depth estimation algorithms. We correct this reference in the revised version.
>
> ### **Point 4: Availability of Source Code:**
> We agree that open-sourcing our implementation will enhance the credibility and reproducibility of our work. We will make the full codebase publicly available upon acceptance of the paper.
>
> Thank you again for your positive feedback and insightful review.

---

### Review · Reviewer_FTjQ · 2025-06-13

**Summary Of Contributions:**

This work proposes a multimodal camera-radar depth estimation setup, with inclusion of an extra 'text' modality to add additional supervision. By leveraging this extra modality, with cues to describe the scene in text, they claim that it improves depth estimation performance.

The architecture is somewhat detailed, but sensible. They generally follow existing ideas for radar camera fusion. For text, they present the image (scene) to a processing network which outputs a text description of the scene. This description is generated according to a prompt, which asks to describe the scene, together with an estimation of the depth. This text is then encoded into text features, processed with an LSTM and fused with radar.

In various places, cross attention is used for fusing modalities. There is a weather prediction head whose output is used in the decoder, taking into account information from all the modalities (radar is noteworthy in adverse weather).

Results
Evaluations are carried out on nuscenes and kitti, showing that their approach works. The ablations show the importance of adding multiple modalities.

**Audience:**

Yes

**Claims And Evidence:**

Yes

**Requested Changes:**

See above. I would like justification for why we enrich text with radar. For instance, why not enrich camera also with it.
In nuscenes, we don't have many adverse weather scenes (as far as I know). How do the adverse weather fittings of this network help here?
In some other work (e.g. RC-PDA), it is seen that radar height measurements from nuscenes sensors are not reliable. It necessitated an elaborate set of operations to clean up this mismatch. Do the authors use any such preprocessing or cleanups in this work? The point I would like to understand is whether the need for such cleanups is obviated in the present work because of evolutions in architecture (I would think not).

**Strengths And Weaknesses:**

This is a principled way to incorporate multimodal fusion with text. The fact that text supervision would contribute in a material way - especially, as it shows up as additional supervision makes sense. I am totally on board with the idea that a more granular set of cues adds richness to the scene rather than simply using bounding boxes or class info.

Some of the design choices seem a bit arbitrary (perhaps that's too strong a statement), such as why would we want to combine text and radar?

How can we expect the text captioning network to predict depth? Why would we trust this?

In the setup, we predict text and then encode it as features (using an LSTM). But why not just take these features from the captioning network instead of transcribing, and encoding them all over again?

---

> ### Author Response · Authors · 2025-06-18
> **Response to Reviewer FTjQ**
>
> We thank the reviewer for observing that incorporating rich textual cues can enhance scene understanding. Below, we address each of your questions in turn.
>
> ### **Rationale for Fusing Text and Radar:**
> Radar sensors offer robustness under adverse conditions (e.g., rain, low light) that can severely degrade image quality. When images are corrupted by noise or poor illumination, automatically generated text descriptions become unreliable or nonsensical. By injecting radar-derived features into the text branch, we supply complementary, modality-agnostic information that stabilizes and enriches the textual representation, thereby improving downstream depth estimation.
>
> ### **Motivation for Text-Augmented Depth Estimation:**
> Vision-language models (VLMs) excel at comprehensive scene understanding, enabling our network with a more human-like perception. Just as people infer depth from a single viewpoint by leveraging prior knowledge of object sizes and their spatial relationships, our model uses richly detailed textual descriptions to form similar priors. In contrast, traditional object detectors output only rigid bounding boxes for a fixed set of categories, and simple captions lack any notion of inter-object context or scene structure.
>
> ### **Text Encoding Pipeline:**
> As detailed in Section 3.1, we partition each image into four quadrants, plus a global view, producing five text paragraphs. We encode each sentence with a pretrained, frozen CLIP text encoder [1], yielding variable-length sets of sentence embeddings per paragraph. To produce a fixed-size feature for each paragraph, we apply an LSTM-based paragraph encoder (Figure 3) over the sentence embeddings. This two-stage process (CLIP → LSTM) ensures a consistent feature dimensionality across frames while preserving both local (region-specific) and global textual information.
>
> ### **Evaluation under Adverse Weather in nuScenes:**
> The nuScenes dataset categorizes scenes into three weather regimes: normal, rainy, and night-time. We observe that nighttime imagery, characterized by motion blur and uneven illumination, poses the greatest challenge for pure visual methods. To improve robustness across all conditions, we introduce a weather-aware fusion block that adaptively integrates radar signals with visual and textual features. In our revised manuscript, we relocate the per-condition performance results (formerly Tables 10–12 in the Appendix, now Table 9 in the main paper) into the main text to highlight these gains.
>
> ### **Characteristics of nuScenes Radar Data:**
> nuScenes radar readings are inherently noisy and lack accurate elevation measurements. Early radar–camera depth pipelines addressed this by first “denoising” radar returns to estimate depth, then fusing with images (RC-PDA [2], RadarNet [3]). More recent work (GET-UP [4], Sparse beats dense [5]) demonstrates that end-to-end learning can directly accommodate sparse, noisy radar without explicit cleanup. Following this trend, our approach operates on raw radar returns, relying on learned feature encoders and fusion blocks to extract and integrate useful depth cues without a separate denoising stage.
>
> [1] Learning transferable visual models from natural language supervision. In International conference on machine learning, pp. 8748–8763. PMLR, 2021.
>
> [2] Radar-camera pixel depth association for depth completion. In Proceedings of the IEEE/CVF Conference on Computer Vision and Pattern Recognition
>
> [3] Depth estimation from camera image and mmwave radar point cloud. In Proceedings of the IEEE/CVF Conference on Computer Vision and Pattern Recognition
>
> [4] Get-up: Geometric- aware depth estimation with radar points upsampling. In 2025 IEEE/CVF Winter Conference on Appli- cations of Computer Vision (WACV),
>
> [5] Sparse beats dense: Rethinking supervision in radar-camera depth completion, 2024. URL https://arxiv.org/abs/2312.00844.

---

### Comment · Action_Editor_esbo · 2025-08-08

The reviewers all recommend acceptance, finding the approach "principled", supported by a "comprehensive" evaluation, showing "superior performance" over reasonable baselines. The review and rebuttal period appears to have been productive as well, addressing a wide variety of minor complaints, and improving the paper. The AE recommends acceptance without further review necessary, but the authors are encouraged to revise the LaTeX carefully to make good use of the space, and ensure that all tables are well-formatted and readable (e.g., Table 9 stands out as unreadable in the current draft).

---

### Decision · Action_Editor_esbo · 2025-08-08

**Recommendation:** Accept as is

**Additional Comments:**

The reviewers all recommend acceptance, finding the approach "principled", supported by a "comprehensive" evaluation, showing "superior performance" over reasonable baselines. The review and rebuttal period appears to have been productive as well, addressing a wide variety of minor complaints, and improving the paper. The AE recommends acceptance without further review necessary, but the authors are encouraged to revise the LaTeX carefully to make good use of the space, and ensure that all tables are well-formatted and readable (e.g., Table 9 stands out as unreadable in the current draft).

**Audience:**

Yes

**Audience Explanation:**

Outdoor depth estimation is a critical problem for embodied agents including cars and robots. Building robustness to adverse weather conditions is a high priority, and this paper makes progress here through creative means, bringing text as a new additional modality.

**Claims And Evidence:**

Yes

**Claims Explanation:**

Good experiments on appropriate datasets, and ablations as well.